# Statistical regularization for trend detection: An integrated approach for detecting long-term trends from sparse tropospheric ozone profiles

Kai-Lan Chang[1, 2], Owen R. Cooper[1, 2], Audrey Gaudel[1, 2], Irina Petropavlovskikh[1, 3], and Valérie Thouret[4]

[1]Cooperative Institute for Research in Environmental Sciences, University of Colorado, Boulder, CO, USA
[2]NOAA Chemical Sciences Laboratory, Boulder, CO, USA
[3]NOAA Global Monitoring Laboratory, Boulder, CO, USA
[4]Laboratoire d'Aérologie, Université de Toulouse, CNRS, UPS, Toulouse, France

**Correspondence:** Kai-Lan Chang (kai-lan.chang@noaa.gov)

**Abstract.** Detecting a tropospheric ozone trend from sparsely sampled ozonesonde profiles (typically once per week) is challenging due to the short-lived anomalies in the time series resulting from ozone's high temporal variability. To enhance trend detection we have developed a sophisticated statistical approach that utilizes a geoadditive model to assess ozone variability across a time series of vertical profiles. Treating the profile time series as a set of individual time series on discrete pressure surfaces, a class of smoothing spline ANOVA (analysis of variance) models is used for the purpose of jointly modeling multiple correlated time series (on separate pressure surfaces) by their associated seasonal and interannual variabilities. This integrated fit method filters out the unstructured variation through a statistical regularization (i.e. a roughness penalty), by taking advantage of the additional correlated data points available on the pressure surfaces above and below the surface of interest. We have applied this technique to the trend analysis of the vertically correlated time series of tropospheric ozone observations from 1) IAGOS (In-service Aircraft for a Global Observing System) commercial aircraft profiles above Europe and China over 1994-2017, and 2) NOAA GML's (Global Monitoring Laboratory) ozonesonde records at Hilo, Hawaii (1982-2018) and Trinidad Head, California (1998-2018). We illustrate the ability of this technique to detect a consistent trend estimate, and its effectiveness for reducing the associated uncertainty in the profile data due to low sampling frequency. We also conducted a sensitivity analysis of frequent IAGOS profiles above Europe (approximately 120 profiles per month) to determine how many profiles in a month are required for reliable long-term trend detection. When ignoring the vertical correlation we found that a typical sampling strategy of 4 profiles-per-month results in 7% of sampled trends falling outside the 2-sigma uncertainty interval derived from the full data set, with associated 10% of mean absolute percentage error. Based on a series of sensitivity studies we determined optimal sampling frequencies for, 1) basic trend detection, and 2) accurate quantification of the trend. When applying the integrated fit method we find that a typical sampling frequency of 4 profiles per month is adequate for basic trend detection, however accurate quantification of the trend requires 14 profiles per month. Accurate trend quantification can be achieved with only 10 profiles per month if a regular sampling frequency is applied. In contrast, the standard separated fit method, which ignores the vertical correlation between pressure surfaces, requires 8 profiles per month for basic trend detec-

tion, and 18 profiles per month for accurate trend quantification. While our method improves trend detection from sparse data sets, the key to substantially reducing the uncertainty is to increase the sampling frequency.

## 1 Introduction

The vertical profile is a type of time series data that reports the composition (e.g. ozone, water vapor, carbon monoxide) or
thermodynamic properties (e.g. temperature, relative humidity, wind speed) of the atmosphere from the surface to an altitude that can range from a few tens of meters (e.g tethered weather balloons) to more than 30 km (e.g. ozonesondes). Conventionally, the analysis of trends based on vertical profile data is conducted on particular altitude bins or pressure levels which are treated as independent time series (e.g., Miller et al. (2006); Harris et al. (2015); Lossow et al. (2019); Petropavlovskikh et al. (2019)), but due to the low sampling frequency of vertical profiles the vertically distributed trend estimates may be inconsistent from
one layer to the next. Trace gas or thermodynamic properties on adjacent pressure or altitude levels are often correlated and the treatment of the levels as independent time series ignores the potential vertical interconnection. In order to reduce the estimation uncertainty of a trend on a particular level, methods have been developed that produce an area average from multiple time series in close spatial proximity (Wigley et al., 1984; Chang et al., 2017), or that employ a dimension reduction technique (e.g. principal component analyses) to project multiple layers of data into a reduced set (Ghil et al., 2002; Meiring, 2007; Park
et al., 2013).

In terms of producing an area average based on multiple stations dispersed across a given region, a typical procedure is to create a surface gridded product, which usually aggregates all available monitoring stations within the grid cell without considering spatial sampling and irregularities (Simmons et al., 2010); another example is trend analysis of ozonesonde profiles, based on units of partial pressure with observations integrated into a limited number of layers for the simplification of analysis
and calculation of column sum (Tiao et al., 1986; Miller et al., 2006). The dimension reduction technique, e.g. principal component analysis, also known as empirical orthogonal function, is commonly used to reduce correlated multivariate data into uncorrelated vectors that maximize the explained variance with as few vectors as possible. This approach is used to calculate certain climate indices, such as Atlantic and Antarctic Oscillations (https://psl.noaa.gov/data/climateindices/list/). However, this technique is a purely numerical algorithm that is not based on physical principles, thus a meaningful interpretation of the
outcome can be subjective or prohibitive.

The aim of such aggregations, either through averaging or dimension reduction, is to enhance a certain common underlying signal (or signals), or to produce regional means (Wigley et al., 1984; Ghil et al., 2002). While the issue of spatial irregularity can now be tackled by a wide range of sophisticated spatial interpolation techniques (Stein, 1999), the averaging approach has not been thoroughly investigated in an objective manner with respect to enhancing the signal to noise ratio. In terms
of calculating long-term trends from sparse time series comprised of vertical ozone profiles, this study seeks a statistical framework to accommodate a broad set of correlations, rather than trying to maximize the signal-to-noise ratio (e.g. trend value and its 2-sigma uncertainty) by deliberately selecting a subset of correlated time series. Specifically, this paper aims to extend the traditional multivariate linear regression model by applying a smoothing spline ANOVA (analysis of variance)

framework (Wang, 1998; Gu, 2004, 2013), to account for the vertical profile variability without a dimension reduction, therefore a physical meaning in each pressure layer could be maintained. A smoothing spline is a non-linear curve or surface fitting technique commonly used to model longitudinal data (for example a large cohort study in biostatistics) (Wahba et al., 1995; Verbyla et al., 1999; Shaddick and Zidek, 2015), and it has been widely applied to a variety of spatial pattern and/or temporal

trend studies (Schlink et al., 2006; Augustin et al., 2009; Park et al., 2013; Wood et al., 2015; Chang et al., 2017). We thus propose a geostatistical approach to fit the vertical profiles jointly as a smooth field for understanding nonlinear interactions and to determine the long-term vertical variability.

Our method uses a two-dimensional geoadditive model for carrying out the multiple correlated time series analysis on every level of the vertical profiles. This approach borrows the strength of spatial correlation models, but instead of working on

dimensions of latitude and longitude, we replace the geographical coordinates with a temporal index and altitude bins. This approach also enables us to identify the common structures among correlated time series (e.g. ozone observations on adjacent layers of the atmosphere) through their seasonality and interannual variability, and to filter out unstructured variations from the underlying signal by a regularization technique. We demonstrate that this method reduces the short-lived anomalies in an irregular and/or sparse time series by taking advantage of the additional information in adjacent layers of the vertical profiles,

and yields a trend with reduced uncertainty. While our method was developed for tropospheric ozone profiles it can also be applied to the stratosphere, profiles of other trace gases or atmospheric thermodynamic properties, or to profiles of oceanic or geologic properties.

Section 2 begins with a brief introduction of the time series decomposition and smoothing spline ANOVA framework, we then propose a statistical methodology to assess the profile variability. Section 3 applies this methodology to two tropo-

spheric ozone profile datasets measured by 1) IAGOS (In-service Aircraft for a Global Observing System) commercial aircraft above Europe and China (including a probability test of how the trend precision and accuracy change according to sampling frequency); and 2) ozonesondes launched from Hilo, Hawaii and Trinidad Head, California, by NOAA's Global Monitoring Laboratory (GML). In Section 4, we discuss the potential extension and benefit of this trend technique, and provide a summary of our ozone profile trend analysis.

## 2  Model and data

### 2.1  Smoothing spline decomposition

Conventional long-term trend detection of a time series is influenced by many factors such as the number of years of data, the magnitude of the uncertainty and autocorrelation of the residuals (Weatherhead et al., 1998). We consider a long-term time series where observations are averaged into monthly means over a number of years in discrete altitude bins or pressure levels.

The basic statistical model for an individual time series can be written as:

$$\boldsymbol{y}(t) = \text{seasonal cycle} + \text{trend} + \text{residual}. \tag{1}$$

where $\boldsymbol{y}(t)$ is the monthly mean ozone value at time $t$. A regular seasonal cycle can be estimated by Fourier harmonic series, cyclic spline functions or by simply averaging the monthly values from the long-term climatology. The specification of a trend component is, however, dependent on the goals of the analysis, and it must be flexible enough to represent the underlying structure in the data (e.g. Augustin et al. (2009); Park et al. (2013); Wood et al. (2015); Chang et al. (2017)); the residual series is generally assumed to be an autoregressive process in this scenario.

Ozone variations are in general strongly dependent on altitude or pressure level. For a given altitude or pressure level, our method borrows the information from the neighboring altitudes, at approximately the same time, for a more stable trend estimate (e.g. observations at nearby altitudes are more highly correlated with the layer of interest than those further away). In order to explicitly account for common structures among correlated time series, a class of generalized additive models (GAMs, Hastie and Tibshirani, 1990) for the functional decomposition can form the general framework for joint trend estimates. Additive models extend traditional multivariate linear models by replacing linear covariates with smooth functions, each represented by various types of spline functions (depending on the nature of the covariate) (Wahba, 1990; Wood, 2006; Wood et al., 2015; Chang and Guillas, 2019). Nonlinear variability and complex interactions can thus be modeled by using the combination of multiple spline basis representations within the GAM framework.

With an extension of a single time series to a vertical profile series (let $\boldsymbol{y}(t,h)$ be the monthly mean ozone value at altitude or pressure level $h$ and time $t$), the statistical model can be extended to a two dimensional signal decomposition:

$$\boldsymbol{y}(t,h) = \boldsymbol{f}_{\text{seasonal}}(\omega,h) + \boldsymbol{f}_{\text{interannual}}(\tau,h) + \epsilon, \tag{2}$$

where

$\omega = 1,\ldots,12$ monthly index;

$\tau = 1,\ldots,n_y$ interannual index (period of coverage);

$t = \omega \times \tau = 1,\ldots,12 \times n_y$ total temporal index;

$h = 1,\ldots,n_h$ altitude bin or pressure level,

where the temporal index $t$ is decomposed into monthly index $\omega$ and interannual index $\tau$, and $\boldsymbol{f}_{\text{seasonal}}(\omega,h)$ and $\boldsymbol{f}_{\text{interannual}}(\tau,h)$ are two dimensional smooth functions representing the seasonal and interannual ozone variations across different altitudes. Specifically, $\boldsymbol{f}_{\text{seasonal}}(\omega,h)$ represents a smooth matrix including a single mean for every month and every altitude (each cross section is an annual cycle for the corresponding altitude), and $\boldsymbol{f}_{\text{interannual}}(\tau,h)$ can be seen as a deseasonalized annual mean anomaly series at each altitude, representing the year-to-year variability. The theory of smoothing spline ANOVA is to decompose a multivariate function into a sum of multiple independent functions with main components and all possible two way (or higher) interactions (up to as many variables as required) (Wang, 1998; Gu, 2004, 2013), in order to understand the contribution from each single component. In terms of our trend study, we only consider the vertical distribution of two basic elements in the time series: seasonal and interannual variability. A higher order of interactions can be included, but it might not be helpful or informative, and could require a substantial amount of computational complexity.

The two dimensional penalized regression splines were chosen for representing both smooth components (Wood, 2003), which can be expressed through a basis representation:

$$\boldsymbol{f}_{\text{seasonal}}(\omega, h) = \sum_{k_1=1}^{K_1} c_{k_1} \boldsymbol{\psi}_{k_1}(\omega, h) \text{ and}$$

$$\boldsymbol{f}_{\text{interannual}}(\tau, h) = \sum_{k_2=1}^{K_2} c_{k_2} \boldsymbol{\psi}_{k_2}(\tau, h),$$

where $\{\boldsymbol{\psi}_{k_1}(\omega, h)\}$ and $\{\boldsymbol{\psi}_{k_2}(\tau, h)\}$ are the collections of smoothing spline basis functions evaluated on the seasonal grid $(\omega, h)$ or the interannual grid $(\tau, h)$, and $\{c_{k_1}\}$ and $\{c_{k_2}\}$ are the collections of associated coefficients (including an overall intercept). Under this formulation we account for temporal and vertical ozone variation in a spatial correlation model (i.e., ozone with different altitudes and different time lags can be correlated, which are not handled simultaneously by the standard multivariate linear regression model). Technical details of the statistical fitting procedure are described in Appendix A.

The specification of our model enables us to decompose the vertical profile into two geoadditive fields: seasonal and interannual components, according to their associated variabilities. These surfaces are defined by approximating the overall monthly means and annual mean anomalies in each layer or altitude through a statistical optimization. Rather than directly specifying a linear component into a multivariate linear regression model, our functional approach aims to partition monthly mean ozone profiles into components attributable to different sources of variation before any attempt at trend detection.

We choose the thin-plate smoothing spline as the spatial basis function because it is computationally efficient, and because it avoids the problem of choosing "knot locations" (Wood, 2006). Note that the focus of this study is jointly modeling multiple correlated time series at different heights above the same region, which differs from a trend study of a series of measurements with uneven temporal sampling at a single altitude (Tiao et al., 1990; Weatherhead et al., 2017), or irregularly distributed observations from a monitoring network (Stein, 1999; Chang et al., 2015, 2017). Our problem can be seen as decomposing

a two dimensional signal according to the associated variability on a regular grid cell (i.e., based on the monthly and annual index at a fixed altitude bin). Specifically, we implemented the penalized regression splines decomposition using the thin-plate splines available from the R package *mgcv* (Wood, 2006; R Core Team, 2013).

## 2.2   Ozone profile observations

Our new trend calculation methodology is applied to the long-term tropospheric ozone vertical profile datasets measured by
1) IAGOS commercial aircraft above Western Europe and China, at 50 hPa intervals from 950 hPa to 250 hPa for the period 1994-2017; and 2) ozonesondes launched from Hilo, Hawaii (1982-2018) and Trinidad Head, California (1998-2018), with a vertical resolution of 100 meters from the surface to 15 km. For consistency the vertical coordinate of the ozonesondes is converted from meters to pressure levels.

      The IAGOS program has measured ozone worldwide since 1994, using instruments onboard commercial aircraft of inter-
nationally operating airlines (Marenco et al., 1998; Petzold et al., 2015, http://www.iagos.org). This analysis utilizes the same subset of IAGOS ozone profiles above Western Europe reported by Cooper et al. (2020). Ozone is measured using a dual-beam

UV-absorption monitor (time resolution of 4 seconds) with an accuracy estimated at about $\pm$ (2 nmol mol$^{-1}$ + 2 %) (Thouret et al., 1998; Nédélec et al., 2015). Because most IAGOS aircraft have belonged to airlines based in Europe since the program began in 1994, Western Europe is the most frequently profiled region in the world. Above Western Europe ($0° - 15°$E, $47° - 55°$ N) IAGOS aircraft measured 34,600 ozone profiles between 1994 and 2016, with 99% of profiles from Frankfurt, Paris, Munich, Brussels, Dusseldorf and Amsterdam. The 22-year time series has one data gap spanning March-August 2010 when instrument failures resulted in just a few sporadic profiles (Petetin et al., 2016b) (the 2010 annual mean is represented by the other 6 months, after adjustments for seasonality). The lower tropospheric portions of the profiles have been shown to be regionally representative of ozone across Western Europe (Petetin et al., 2018). The sampling frequency varies according to airline schedules, but on average, four profiles are recorded somewhere in this region every day. IAGOS aircraft can take-off and land at any time of day and all data are used in this analysis. No diurnal ozone cycle occurs in the free troposphere above Europe (above the 750 hPa level), although a clear ozone cycle occurs in the boundary layer, and is strongest below 950 hPa (Petetin et al., 2016a). Our analysis only calculates ozone trends on the 950 hPa surface and above, which avoids the lowest layers with very strong diurnal cycles. We analyze a similar IAGOS data set based on a compilation of ozone profiles from airports in northeastern China, and Seoul, South Korea. The Europe and China datasets focus on limited regions of central-Western Europe (0-15°E, 47-55°N)) and northeast China/South Korea (110-135°E, 28-45°N).

NOAA's Global Monitoring Laboratory has measured ozone profiles above Hilo, Hawaii (19.72°N and 155.05°W) since 1982, and Trinidad Head, California (41.06°N and 124.15°W) since 1997, using balloon-borne ozonesondes equipped with electrochemical concentration cell (ECC) sensors that have an accuracy of $\pm 10\%$ in the troposphere (Johnson et al., 2002; Smit et al., 2007). The ozonesondes have been launched on a weekly schedule since 1992 (Oltmans et al., 1996) with a lower frequency of approximately 2-4 profiles per month from 1982 to 1991. Compared to UV-absorption measurements, ECC ozonesondes show a modest ($\sim$1-5% $\pm$ 5%) high bias in the troposphere, but no change over time (Tarasick et al., 2019). The Hilo ozonesonde record has been reprocessed according to the recommendations of the Ozonesonde Data Quality Assessment activity (Smit et al., 2012; Smit, H. G. J. and Panel for the Assessment of Standard Operating Procedures for Ozonesondes (ASOPOS), 2014; Sterling et al., 2018), to remove artifacts introduced by changes of sonde type, manufacturer, strength of sensing solution, or preparation procedure. The sampling frequency at Trinidad Head is approximately once per week since August 1997.

## 3  Results

The IAGOS dataset in western Europe is a regional aggregation of profiles from several airports. The profiles are measured by several aircraft and on average there are approximately four profiles per day within the study region. Several studies have demonstrated that IAGOS data above Western Europe are consistent with ozonesonde records in the UTLS (Staufer et al., 2013, 2014), and the data have compared well to regional surface and free tropospheric ozonesonde records (Thouret et al., 1998; Logan et al., 2012; Petetin et al., 2018). Petetin et al. (2018) validated IAGOS data in the lowest troposphere over Western Europe against rural monitoring sites at various elevations and concluded that those observations can be used to study the air

quality in the agglomeration. Nevertheless, to reduce the influence from the strong diurnal cycle at the surface, all observations below the 950 hPa level and within 300 m of the surface were removed from the analysis (Gaudel et al., 2020).

With such a high sampling frequency, the data are expected to be far less inconsistent than a sparsely sampled ozonesonde time series (one to three profiles per week), and highly representative of the true ozone variability above Western Europe. The

IAGOS dataset in China is also a regional aggregation, but the time series has a period of low sampling frequency (2007-2010), with many months containing no data. In contrast, the ozonesonde records at Hilo and Trinidad Head are expected to have greater uncertainty, due to the lower sampling rate of one profile per week. Our first task was an evaluation of our method against the standard linear model using the IAGOS data above Europe, to determine if our method yields trend values with reduced uncertainty. The next step was to examine the impact of sampling frequency on the trend estimate, based on the IAGOS

profiles above Western Europe. Finally, we applied our method to the sparsely sampled Hilo and Trinidad Head ozonesonde records to demonstrate the improved trend quantification.

## 3.1   IAGOS trends above Western Europe

Figure 1 shows the pairwise simple correlation coefficients of IAGOS data between every 50 hPa layer from from 950-250 hPa, based on all de-seasonalized monthly ozone values from 1994 to 2017. As expected from previous work (Petetin et al.,

2016b), the results show high correlation between adjacent layers, which decays with distance. Time series are well correlated ($\sim 0.6 - 0.9$) within a vertical range of 100 hPa, except for the top layer where correlations decrease with distance at twice the rate of the lowest layers. As noted by Weatherhead et al. (2017), two time series can be highly correlated even if they have very different mean values or magnitudes of variability. Because our technique takes advantage of the nearby layers with moderate or high correlation, we expect an improved quantification of the absolute variability and resulting trend estimate.

The smooth seasonal and interannual components in units of ppbv (parts per billion by volume) derived from Eq (2) are shown in Figure 2(a) and (b), respectively. For each component the bottom panel provides the one dimensional climatological seasonal cycle or the annual estimated mean anomalies on each pressure layer, while the top panel shows their continuous fields. The combination of the two fields represents our overall fit to the vertical profile data. For example, the fitted result for the year 2000 is equal to the sum of the column values (by pressure) at 2000 in Figure 2(b), and the corresponding column

values in Figure 2(a). In addition, based on the variogram fitting to the interannual component (Stein, 1999), two ozone time series separated beyond 150 hPa are no longer (auto-)correlated.

From a visual inspection of the interannual component, the variability in the lower layers is fairly steady and unwavering compared to the upper layers. The largest variation over 1994-2017 occurred in the upper layers around 2017 (the absolute magnitude of the variability during 2017 is still in question as it was the last year of the available time series, and could be

modified by additional years of data as they become available). ~~A comparison of the magnitudes of the seasonal and interannual distributions suggests that the seasonality remains the principal source of the tropospheric ozone variation, as expected from many previous ozone trend studies.~~ The result from the smoothing spline decomposition provides a good visualization of the trend, and also provides a quantification of ozone variation through the multiple correlated time series, using all profile data without a dimension reduction.

Figure 3 provides some statistical model diagnostics for the overall fit. In general the model fits the data well, but heteroscedasticity is present, as expected. The model underestimates the extreme values, which is deliberate because we developed this penalized regression approach to prevent overfitting. Figure 3b-d presents the marginal residual distributions by year, month and pressure layer. Overall, we can see that the mean structures of seasonal, interannual and vertical variations are well

represented (the medians of residuals in the boxplots are close to zero and the interquartile ranges are well constrained), but unsurprisingly the variance structures are not completely captured by our model. Heteroscedasticity is most prominent in the top pressure layers, certain year such as 2017, and the sub-seasonal variability. Besides the extreme values, the diagnostics indicate that an extended model with a more complicated variance component, or additional climatic index for a specific layer or month/year, is required for explaining the remaining variability. While the model cannot explain all of the variance, applying

our method to long-term, multi-decadal records reduces the impact of the unstructured variations on the trend estimate (as illustrated below).

Using the interannual component in Figure 2(b) we calculate the trend value for each individual pressure layer. This integrated fit is then compared to the separated linear fit in each layer based on Eq (1) and the original monthly mean time series (with seasonal cycle and autocorrelation accounted for). Note that the trend result from the integrated fit (i.e. via the smoothing spline ANOVA model) takes advantage of the correlated observations from neighboring layers, thereby reducing the unstruc-

15 tured variation in the data set, while the separated fit is obtained by applying a simple linear fit to the original (and rather inconsistent) time series on each independent pressure level.

As shown in Figure 4, these two methods agree very well, because, as shown in Section 3.2, the monthly means are generated from very large sample sizes (approximately 120 profiles each month), which minimizes the uncertainty on the simple linear

trend in any given layer. A slight discrepancy can be seen above 400 hPa; to investigate the cause we plot the observed time series and model fitted values for 4 pressure levels in Figure 5. The year-to-year variations at 400 hPa are well captured by the model fit. Above 350 hPa, a large spike in the observed values occurred in 2017, allowing us to see how the penalized regression spline responds to such sharp variations. Especially at 250 hPa, if only one month shows a spike, such as in 2000, 2001, 2006, 2008 and 2013 (and if these spikes are not obvious on adjacent layers, such as at 300 and 350 hPa), the variation will be

penalized and smoothed out. On the other hand, if the enhancement occurs over multiple months, then it will be reflected by the model, such as 2017. These results demonstrate that the smoothing spline decomposition is a powerful tool for identifying and replicating persistent features in the ozone profiles, while ignoring short-lived anomalies.

To demonstrate the role of regularization in the model fitting, we first provide a synthetic example that illustrates the problems of underfitting (the model is not flexible enough to capture the general pattern in the data) and overfitting (the model is

30 overparameterized, so some unphysical fluctuations are present) in supplementary Figure S1. Neither underfitting nor overfitting is an appropriate representation of the true process. Once sufficient model complexity is supplied (e.g. placing a knot for spline functions every 50 hPa), the statistical regularization can be used to penalize overly complex models and thus prevent overfitting. The overfitting of a surface is less obvious than a curve, but we provide a demonstration of the fitting by adjusting the optimized penalty coefficient (which is selected by the generalized cross validation (Wood, 2006) and is proven to be reli-

35 able, as discussed above). We scale the penalty coefficient ($\lambda_2$ in Equation (A2) while keeping $\lambda_1$ fixed) by a factor of 10 and

0.1, respectively (i.e. the smaller penalty, the more roughness will be present). The result is shown in Figure 6: the underfitting can be seen as an over-smoothed representation of the underlying structure, and sharp variations (e.g. overly complicated roughness) are an indication of overfitting.

Figure 4 shows that the ozone trends increase with altitude, ranging from 3-8 ppbv decade$^{-1}$ above 400 hPa, as determined by the integrated fit method. These strong trends are driven by the ozone in the stratospheric air masses that are frequently sampled at these altitudes. To focus on the ozone trends in the upper tropospheric air masses we removed all of the stratospheric air mass samples from the time series, as determined from the potential vorticity values associated with each ozone profile, and available from the IAGOS data portal (https://doi.org/10.25326/20) (Cohen et al., 2018). Removal of the stratospheric air masses reduces the monthly mean ozone values above 400 hPa by approximately 10 ppbv, and reveals a much more subtle year-to-year variability, especially in the upper levels (supplementary Figure S2). Overall the trends in the upper levels are reduced by more than a factor of two when the stratospheric air masses are removed, but they are still greater than the trends in the mid-troposphere (supplementary Figure S3).

## 3.2 Effect of sampling frequency on IAGOS trends above Europe

Due to ozone's high temporal variability, accurate trend detection may not be possible if the ozone profile sampling rate is low. Three previous studies have estimated the optimal sampling frequency for accurate quantification of tropospheric ozone variations from profiles. Logan (1999) concluded that at least 20 profiles per month are required in tropical and extratropical regions to derive reliable monthly mean ozone values. Saunois et al. (2012) analyzed free-tropospheric IAGOS profiles above Frankfurt (1995-2008) and determined that sampling frequencies of 4 and 12 profiles per month approximately result in uncertainties of 9-29% and 5-15% on the seasonal mean, respectively. Leonard et al. (2017) determined that at least 3-6 ozonesonde profiles per month are required for the estimation of climatological ozone values, in terms of matching the 95% confidence level of the fully sampled monthly ozone means derived by the GFDL-AM3 model.

Here we determine the optimal ozone profile sampling frequency for trends calculated using either the separated fit or the integrated fit methods. In order to investigate the impact of sampling frequency on profile trends, we randomly selected a specified number of profiles per month (from 1 to 20) and conducted the same integrated fit analysis on the resulting monthly mean time series, based on 1000 iterations of random sampling. We illustrate the impact of the sampling frequency by showing the trend results based on 1, 5 and 9 profiles per month in Figure 7. The plot shows the vertical distributions of sampled trends by separated and integrated fits. We also show the histogram of sampled trend values at 550 hPa (where the trends and 2-sigma intervals are quite similar for these two approaches), and associated trend fitting uncertainties in Figure 8. We can see substantial improvement of the trend estimate when more data are available, with a similar trend accuracy for both approaches. However, in terms of trend precision, the integrated fit produces lower uncertainty.

If we assume that the trend derived from the full data set (dozens of profiles per month) is the "true" value, the sampled trend is randomly located around the true value as a Gaussian distribution for both approaches. For both the integrated and separated fit methods, the increased sampling frequency results in a diminished distribution of trends. The integrated fit is designed to smooth out the short-lived anomalies in the vertical profiles, and as a result the integrated fit produces a narrower range of

trend distributions than the simple separated fit. This case study clearly illustrates that an inconsistent trend structure results from infrequent sampling.

The summary statistics for the sensitivity analysis in terms of both precision and accuracy are reported in Table 1. Due to a lack of systematic behavior across the 15 vertical layers between 950 and 250 hPa in this sensitivity analysis, we do not report individual layer results; instead, each number in the table represents an average from 15000 samples (1000 samples per layer). It should be noted that all the interpretations of this analysis are based on two assumptions: (1) time series are long-term, i.e. twenty years or longer, and (2) the underlying variations are fairly steady and regular. Therefore our results indicate the minimal requirement for the sampling scheme.

As an indication of the trend precision, we calculate the percentage of the 15000 random trend values that fall outside of the 1- or 2-sigma intervals of the reference trend, which is based on the full data set. The integrated fit delivers an improved precision as demonstrated by the lower outlier rate (i.e a higher coverage rate) in relation to a narrower uncertainty interval. For the trend accuracy metric, we calculate the mean absolute percentage error (MAPE) for each sampling profile:

$$\text{MAPE} = \frac{100\%}{n_h} \sum_{h=1}^{n_h} \left| \frac{p_h - a_h}{a_h} \right|,$$

where $p_h$ and $a_h$ are the sampled and the assumed true trend values at height $h$, with the averaged metric reported in Table 1. The separated fit method improves the accuracy but to a relatively smaller degree than its improvement on the precision. While our sophisticated statistical method clearly improves the accuracy and precision of the trends, greater improvements can be achieved by simply increasing the sampling rate by one or two profiles per month.

Figure 9 visualizes Table 1 and shows the sampling rates that allow the precision and accuracy to approach that of the complete data set. At least 4 or 5 profiles per month are required to arrive at a 5% outlier rate with respect to the 2-sigma interval for the integrated or separated fit, respectively. Improved precision becomes negligible at sample sizes greater than 15 profiles per month. In terms of accuracy, a monthly sampling frequency of 14 profiles is required to reduce the error to 5% using the integrated fit, while the separated fit requires 18 profiles per month to meet the same goal.

In order to investigate the impact of sampling strategy on the integrated and separated fits, we carry out an additional sensitivity analysis in supplementary Table S2 by a comparison of two strategies: (A) a completely random design as illustrated above; (B) a fixed sampling strategy based on the random selection of an initial day followed by additional profiles at fixed intervals of 1 to 10 days. For example, a 5-day sampling frequency is based on the random selection of an initial reference from day 1 to day 5 in the beginning of the record, followed by the random selection of a profile every 5th day until the end of the record. For both strategies, only a single profile is chosen randomly if multiple profiles are available on the same day. Therefore the sampling scheme with a fixed frequency of 1 day represents a random selection of a single profile in each day, instead of using all available data. Also, if the chosen day does not have any profiles, we treat it as missing and do not look for a replacement.

The sensitivity analysis demonstrates that sample size remains the dominant influence on precision and accuracy. When the sampling interval is not dense enough (e.g. greater than 10-days), the benefit of a regular frequency scheme is inconsequential. However, when the monthly sample size is greater than 4 profiles (i.e. the sampling frequency is less than once per week),

the fixed frequency scheme could achieve a better performance. As a result, an optimal frequency can decrease to 10 profiles (3-day frequency) for an integrated fit and 15 profiles (2-day frequency) for a separated fit.

The above discussion is focused on the precision and accuracy of the trend estimate at various sampling frequencies. In addition, we can explore the impact of sampling frequency on our ability to simply detect the presence of a trend, based on uncertainty analysis. In order to evaluate the resulting uncertainty associated with the sampled trends, we include the mean signal-to-noise ratio (MSNR) between trends derived from the full data set (the assumed true trend values) and the uncertainty in each sample (i.e. standard error associated with the trend estimate) in Table 1. In the interest of a fair comparison, this calculation is done by comparing the sampling uncertainty with trend values derived by the same method. Note that we do not use the concept of "statistical significance" to indicate evidence for the trends, following the recent recommendations from the American Statistical Association (Wasserstein and Lazar, 2016; Wasserstein et al., 2019). Instead, the benchmark we selected for comparing trend uncertainties is a rejection of the null hypothesis at the 95% confidence when the SNR exceeds a threshold value of 2. As noted in Table 1 we already knew the MSNR is around 2.3 for both methods when using the full datasets, however, the SNR analysis shows that the benchmark value of 2 can be achieved at a sampling frequency of just four profiles per month when using the integrated fit, whereas the separated fit requires 8 profiles per month. In summary, at a low sampling frequency the integrated fit provides not only more precise and accurate trend estimates, but the uncertainty associated with trends can be reduced.

## 3.3 Application of the integrated fit to IAGOS trends above NE China

IAGOS aircraft have measured ozone profiles above northeastern (NE) China and Seoul, South Korea since 1994, but at a much lower sampling rate than Western Europe. While Western Europe has 36298 profiles during 1994-2017, NE China (including profiles from Seoul, South Korea) only has 1636 profiles with many months containing sparse data during 2007-2010. The lower sampling frequency and the data gap mean that the NE China IAGOS dataset can provide a further test of the advantages of the integrated fit method. The seasonal and interannual components of the integrated fit to NE China are shown in Figure 10. Compared to Western Europe the seasonal component appears more incoherent in the vertical, partially due to a lower sampling frequency, however a clear seasonal peak can be seen in June in the upper troposphere and in April-May-June in the lower troposphere. The interannual component shows a steady increase of ozone at all pressure layers from 1994 to 2005, but rather large fluctuations after 2005, in both the upper and lower troposphere.

In this particular region, the variogram fitting suggests that ozone time series within a range of 200 hPa should be considered as correlated. This vertical correlation range is greater than was found for the IAGOS data above Western Europe. The longer correlation range is the result of the more systematic temporal variations across ozone vertical profiles, as seen in Figure 10(b). While the correlation structure above Europe is heavily affected by the high anomalies from stratospheric influences, if those high anomalies are filtered out (see supplementary Figure S2), the vertical correlation range increases and becomes similar to the IAGOS data above China. Time series of observations and fitted values for four upper and lower layers are provided in supplementary Figures S5 and S6, respectively, to illustrate the statistical fitting quality.

The resulting trend distribution above China is displayed in Figure 11. The separated fit finds that the 2-sigma interval of trend values is inseparable from zero at 600 and 300 hPa, while the integrated fit detects positive trends with lower uncertainties throughout the depth of the troposphere, with a larger trend in the boundary layer and a smaller trend at 500 hPa. Compared to the separated fit, which is a typical example of overfitting to the unstructured variation at a small scale, our statistical adjustment that borrows information from adjacent layers is effective in terms of a better representation between fidelity and complexity.

## 3.4 Application of the integrated fit to the Hilo and Trinidad Head ozonesonde records

Ozonesondes have been launched from Hilo, Hawaii, continuously since 1982, at an average rate of 3 profiles per month for the first 10 years and 4 profiles per month since 1993; the sampling rate at Trinidad Head has been weekly since August 1997. Figure 12 shows all Hilo and Trinidad Head ozone observations by decade, from the surface to 15 km, at 100 m intervals, along with profiles of the 5th, 50th and 95th percentiles in each decade. From a visual inspection, the decadal 50th percentile profiles at Hilo are similar in the lower troposphere, but the profiles diverge above 700 hPa (∼3km), and a clear enhancement of ozone can be seen after the year 2000. The decadal 5th, 50th and 95th percentile profiles at Trinidad Head are similar during the 2000s and 2010s, while the much shorter record over 1997-1999 indicates higher ozone values in the upper troposphere.

The results of the functional decomposition to distinguish the seasonal and interannual components of the Hilo record are shown in Figure 13. The magnitude of the seasonal peak increases with altitude, while the interannual component reveals a wide range of vertical variation. With a nearly flat variation in the lower troposphere and strong fluctuations at higher altitudes, the ability of the smoothing spline decomposition to identify a complex range of variations is clearly illustrated. The same plot for Trinidad Head is displayed in Figure 14. The seasonality varies with altitude, with maximum values occurring in March in the upper troposphere and in May in the lower troposphere. The annual anomalies show greater variability in the upper troposphere, especially in the first four years when a strong positive anomaly abruptly transitioned to a strong negative anomaly. The upper tropospheric enhancement above mid-latitude western North America in 1998 has been reported previously (Langford, 1999; Cooper et al., 2010), and this feature has been attributed to enhanced stratosphere-troposphere exchange following the strong 1997-1998 El Niño event (Langford, 1999; Lin et al., 2015).

Note that increasing the vertical resolution of the profile data does not play an important role in the quantification of the vertical correlation range, as the correlation range of a fitted variogram is still approximately 150-200 hPa in the troposphere for these two stations, as previous results suggested. The vertical distributions of the Hilo (1982-2018) and Trinidad Head (1998-2018) ozone trends from the separated and integrated fits are shown in Figure 15, reported at a 20 hPa vertical resolution, as opposed to 50 hPa resolution for the IAGOS profiles above Europe and China. For reference, the trend estimation from the 1982-2018 surface ozone measurements at Mauna Loa Observatory (∼3400 m elevation; 60 km southwest of Hilo) was added to the Hilo plot (Cooper et al., 2020). The separated fit results in trends that can vary considerably from one layer to the next. For example, from 650 hPa to 350 hPa at Hilo the trend uncertainty range includes zero for some levels, but not for others. In contrast, the integrated fit produces narrower uncertainty ranges because it avoids the influence of extreme events and borrows information from adjacent layers, resulting in all uncertainty ranges exceeding zero above 650 hPa. Supplementary Table S1

reports the trend value and associated 2-sigma uncertainty above Hilo and Trinidad Head for the tropospheric column, along with the partial column in the lower, middle and upper troposphere, for the full records and since a common reference year of 2000.

For the trend distribution at Trinidad Head, the overall trend profile is shifted toward negative values due to the positive ozone anomaly in 1998/1999 (as described above), but above 900 hPa the 2-sigma uncertainty range still does not exclude zero. Supplementary Figure S7 shows the trend profile when the time series begins in 2000, to avoid the positive anomaly in 1998/1999. Since 2000 trends are very weak, except for the boundary layer where trends are generally negative.

## 4    Conclusions

Detecting trends of tropospheric ozone from ozonesonde profiles is challenging due to relatively low sampling frequency combined with high temporal variability. Regularization is a statistical learning tool which makes a trade-off between, 1) high fitted bias (low flexibility) that results from underfitting, and 2) high sensitivity to small data fluctuations (low generalizability) that results from overfitting. The underfitting can be avoided by making sure the number of basis functions (e.g. thin-plate splines) are large enough to represent the underlying process. A model can be considered to be overfitted if a high cross validation error is found (which can be investigated by, e.g., iteratively removing one observation and predicting this value from the remaining observations). In terms of detecting tropospheric ozone trends from vertical profiles, the vertical correlated structures in the neighboring pressure layers can be used to inform the learning process. The benefit of this approach can be reflected by the detectable trends (if any) at a low sampling rate (i.e. we have higher confidence of our ability to detect a trend from weekly sampled ozonesonde data), and by an improved quantification of trend estimates in terms of accuracy and precision.

This technique efficiently reduces the uncertainty of the resulting trends, and thus increases our ability to detect and quantify trends of smaller magnitude. Therefore, this method is valuable for improved trend detection of ozonesonde records, because although these records are sufficiently long-term and have high vertical resolution with high accuracy, standard trend analysis is still challenging due to the limited sampling rate. This refined estimation is also expected to be beneficial when comparing trends derived from different regions or observing systems, since the result is less sensitive to incoherent or unstructured variations.

This paper provides a sophisticated statistical regularization approach for carrying out a joint trend analysis of vertical profile data, as applied to tropospheric ozone observations from IAGOS commercial aircraft profiles above Europe and China, and NOAA GML's ozonesonde records at Hilo, Hawaii and Trinidad Head, California. Our approach is designed to deliver a consistent trend and uncertainty estimate by functionally decomposing the overall vertical profile into seasonal and interannual components. Instead of fitting the trend component as a single linear term, our technique adopts a two-step data driven approach that, 1) expresses the continuous change as a series of estimated annual mean anomalies in each altitude or pressure level, after accounting for deseasonalization and vertical autocorrelation; 2) and then derives the trend value from the annual mean anomalies according to its underlying structure (a linear approximation of the interannual variability is considered to be

appropriate). In this study we introduce roughness penalties on both monthly and annual scales, which are the fundamental elements of time series decomposition. This data-driven approach reveals the underlying trend structure without determining the linear, nonlinear or any other type of polynomial in advance. The success of this approach lies in the smoothing spline decomposition of ozone profile variability. Importantly, the regularization aspect diminishes the influence from extreme observations. Our method is easy to implement under low computational costs, and it offers a spatial visualization for investigating the trend and interannual variability typical of ozone profile data.

While this study focused on temporal and vertical variations at a specific location or region, an extension to a spatial-varying vertical process (i.e. incorporation of multiple sites with dimensions of longitude and latitude) is theoretically possible. However, four-dimensional correlations could be difficult to disentangle, while still maintaining computational tractability. The scaling problem beyond three dimensions could be even more challenging, as it involves multiple measurement units in space and time.

The main results of the ozone trend analysis are summarized as follows:

1. The abundance of IAGOS ozone profiles above Europe is sufficient to provide a reliable trend result without the need for applying an advanced statistical technique. Nevertheless, a sensitivity analysis demonstrates that at low sampling frequencies the integrated fit outperforms the separated fit in terms of accuracy and precision.

2. A further sensitivity test confirms that regular sampling frequency can be more beneficial than random sampling under the same number of available profiles.

3. While our method improves trend detection from sparse data sets, the key to substantially reducing the uncertainty is to increase the sampling frequency.

4. Based on a series of sensitivity studies we determined optimal sampling frequencies for, 1) basic trend detection, 2) and accurate quantification of the trend. When applying the integrated fit method we find that a typical sampling frequency of 4 profiles per month is adequate for basic trend detection, however accurate quantification of the trend requires 14 profiles per month. Accurate trend quantification can be achieved with only 10 profiles per month if a regular sampling frequency is applied. In contrast, the standard separated fit method, which ignores the vertical correlation between pressure surfaces, requires 8 profiles per month for basic trend detection, and 18 profiles per month for accurate trend quantification.

5. Trends derived from the separated and integrated fits are similar above China: stronger positive signals in the upper and lower troposphere, and relatively weaker positive signals in the middle troposphere. However, the integrated fit yielded a more consistent trend estimate.

6. Application of the integrated fit to the Hilo record enables us to effectively reduce the unstructured variation in the profile and derive clear positive trends in the min- and upper troposphere.

7. The reference year of the trend estimate is particularly crucial at Trinidad Head, due to high anomalies at 1998-1999. The trends are weak in the mid- and lower troposphere, and tend to be more negative in the upper troposphere.

Finally, although this method was developed for tropospheric ozone profiles it can also be applied to the stratosphere, profiles of other trace gases or atmospheric thermodynamic properties, or to profiles of oceanic or geologic properties. We provide an example in Supplementary Figure S8 to illustrate the application of this method to the quantification of trends in the lower stratosphere above Hilo, Hawaii.

*Code and data availability.* The IAGOS data are publicly available at https://doi.org/10.25326/20. The ozonesonde data can be found in the NOAA ESRL Global Monitoring Laboratory (GML) database at ftp://aftp.cmdl.noaa.gov/data/ozwv/Ozonesonde/. The R code for the proposed approach is available on GitHub at https://github.com/Kai-LanChang/VerticalProfileStudy.

## Appendix A:  Statistical details

This model proposed in Section 2 can be fit to the data and the coefficients can be estimated by minimizing the penalized least
square criterion (Wood, 2006)

$$\left\| \boldsymbol{y}(\omega, \tau, h) - \boldsymbol{f}_{\text{seasonal}}(\omega, h) - \boldsymbol{f}_{\text{interannual}}(\tau, h) \right\|^2 + D^2 \mathbf{F}. \tag{A1}$$

where $\| \cdot \|$ is the Euclidean norm. The first term in Equation (A1) minimizes the residual sum of squares (quantifying the goodness of fit). The second term is a nonnegative measure of the complexity of $\mathbf{F}$, which penalizes the roughness of the curve or surface (avoiding over-fitting to the short-lived anomalies) defined in terms of an integral of second-order partial derivatives
of $\mathbf{F}$.

On the geometry of a function, the second derivative corresponds to the curvature or concavity of the field, thus this penalty term is designed to prevent a "standalone" sharp variability from its neighborhood system. For example, except for the top and bottom layer, the variability in each layer should be bounded by its upper and lower layers, and any variability exceeding these bounds is potential short-lived anomaly. Specifically, the penalty term in Equation (A1) can be defined as (Wood, 2003):

$$
\begin{aligned}
\quad D^2 \mathbf{F} = {} & \lambda_1 \iint \left[ \frac{\partial^2 \boldsymbol{f}_{\text{seasonal}}(\omega, h)}{\partial \omega^2} \right]^2 + \left[ \frac{\partial^2 \boldsymbol{f}_{\text{seasonal}}(\omega, h)}{\partial h^2} \right]^2 + 2 \left[ \frac{\partial^2 \boldsymbol{f}_{\text{seasonal}}(\omega, h)}{\partial \omega \partial h} \right]^2 \mathrm{d}\omega \mathrm{d}h \\
& + \lambda_2 \iint \left[ \frac{\partial^2 \boldsymbol{f}_{\text{interannual}}(\tau, h)}{\partial \tau^2} \right]^2 + \left[ \frac{\partial^2 \boldsymbol{f}_{\text{interannual}}(\tau, h)}{\partial h^2} \right]^2 + 2 \left[ \frac{\partial^2 \boldsymbol{f}_{\text{interannual}}(\tau, h)}{\partial \tau \partial h} \right]^2 \mathrm{d}\tau \mathrm{d}h,
\end{aligned}
\tag{A2}
$$

where $\{\lambda_1, \lambda_2\}$ are the tuning parameters for seasonal and interannual components, respectively. A major feature of the roughness penalty term in the regression spline approach is the isotropic property, which means that the roughness penalty is invariant in all directions (i.e. across altitude and time). This isotropic property is natural and straightforward when considering smooth
spatial structures on the same geographic coordinates. However, since time and space are measured in different units, we cannot interpret or distinguish the relative importance between smoothness in time and smoothness in space. A pragmatic but efficient approach for avoiding such scaling issues is to transform all variables into the unit hypercube (Sobol, 2001). We considered a potential extension for our particular problem by specifying a different smoothing parameter for each derivative term in Equation (A2) (Wood, 2006), however this step proved ineffective as the specification of many smoothing parameters resulted in an

over-smooth field. The primary concern in this study is to borrow the strength of common variabilities among neighboring time series for a better trend estimate, with one penalty for the monthly scale and another penalty for the annual scale which are reasonable for our application, as long as the relative vertical distance for the different time series is appropriately documented.

The solution for the penalized least square in Equation (A1) is provided by the joint $(K_1 + K_2)$ coefficient vector $\boldsymbol{c} = (c_{k_1}, c_{k_2})^\top$, as follows:

$$\boldsymbol{c} = (\mathbf{X}^\top \mathbf{X} + \lambda_1 \mathbf{S}_1 + \lambda_2 \mathbf{S}_2)^{-1} \mathbf{X}^\top \boldsymbol{y},$$

where $\mathbf{X}$ is a $(12 \times n_y \times n_h) \times (K_1 + K_2)$ covariate matrix that stacks up all smooth basis functions $\{\psi_{k_1}\}$ and $\{\psi_{k_2}\}$, and $\mathbf{S}_1$ and $\mathbf{S}_2$ are known penalized matrices consisting of the smoothing spline basis functions from Equation (A2). If $\lambda_1 = \lambda_2 = 0$, the solution is equivalent to the ordinary least square estimate. Extremely large values of the tuning parameter depreciate the curve features and could lead to a plane estimate for $\mathbf{F}$. The estimation of tuning parameter $\boldsymbol{\lambda} = \{\lambda_1, \lambda_2\}$ can be solved by minimizing the generalized cross validation (GCV) score, which is defined as (Golub et al., 1979):

$$\mathrm{GCV} = \frac{N \|\boldsymbol{y} - \hat{\boldsymbol{y}}\|^2}{[N - \mathrm{tr}(A(\boldsymbol{\lambda}))]^2}, \tag{A3}$$

where $A(\boldsymbol{\lambda}) = \mathbf{X}(\mathbf{X}^\top \mathbf{X} + \lambda_1 \mathbf{S}_1 + \lambda_2 \mathbf{S}_2)^{-1} \mathbf{X}^\top$ is the influence matrix such that $A(\boldsymbol{\lambda})\boldsymbol{y} = \hat{\boldsymbol{y}}$, $\mathrm{tr}(A(\boldsymbol{\lambda}))$ is the trace (sum of the elements on the main diagonal) of $A(\boldsymbol{\lambda})$, and $N = n_h \times n_y \times 12$ is the total number of data points. Furthermore, the number of basis functions has to be selected, which is not a part of the optimization problem. The number of basis functions for each component is adjusted cautiously so that the impact on the result is negligible from a further increase of basis functions (here $K_1 = 120$ and $K_2 = 300$, a full spline representation would be up to $K_1 = n_h \times 12$ and $K_2 = n_h \times n_y$, but it is considered to be computationally wasteful). Further technical details on spline functions in the penalized least square can be found in Wood (2006).

*Author contributions.* KLC and ORC contributed to conception and design. AG, IP and VT contributed to acquisition of data. KLC conducted the analysis. KLC and ORC drafted the article while AG, IP and VT helped with the revision. All authors approved the submitted and revised versions for publication.

*Competing interests.* The authors have no competing interests to declare.

*Acknowledgements.* We thank Toshihiro Kuwayama (Toshihiro.Kuwayama@arb.ca.gov) and the California Air Resources Board for supporting the ozonesonde program at Trinidad Head in recent years, and Michael Ives at Humboldt State University for his many years of collaboration with NOAA GML and his dedication to launching weekly, and sometimes daily, ozonesondes from Trinidad Head. The IAGOS program acknowledges the European Commission for its support of the MOZAIC project (1994-2003) the preparatory phase of IAGOS (2005-2013) and IGAS (2013-2016); the partner institutions of the IAGOS Research Infrastructure (FZJ, DLR, MPI, KIT in Germany,

CNRS, Météo-France, Université Paul Sabatier in France, and University of Manchester, UK); the French Atmospheric Data Center AERIS for hosting the database; and the participating airlines (Lufthansa, Air France, China Airlines, Iberia, Cathay Pacific, Hawaiian Airlines) for transporting the instrumentation free of charge.

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

**Table 1.** Percentage of randomly generated trends that fall outside of the 1- or 2-$\sigma$ intervals of the trend values that were derived from the full IAGOS data set. One thousand randomly generated trends were calculated on 15 pressure surfaces for each of a pre-determined number of profiles per month. Also shown are the associated mean absolute percentage error (MAPE) values, and mean signal-to-noise ratio (MSNR) between trends derived from the full data set and the uncertainty in each sample (i.e. standard error of the trend estimate). The assumed true MSNR values derived from the full data set are 2.32 and 2.34 for separated and integrated fits, respectively.

| Number of profile-a-month | | 1 | 2 | 3 | 4 | 5 | 6 | 7 | 8 | 9 | 10 |
|---|---|---|---|---|---|---|---|---|---|---|---|
| Separated fit | 1-$\sigma$ range [%] | 61.4 | 48.4 | 41.1 | 36.2 | 32.1 | 28.7 | 23.3 | 21.0 | 18.8 | 16.8 |
| | 2-$\sigma$ range [%] | 31.7 | 16.8 | 10.8 | 7.2 | 5.0 | 3.7 | 2.3 | 1.6 | 1.2 | 0.7 |
| | MAPE [%] | 18.4 | 13.2 | 11.3 | 10.1 | 9.2 | 8.5 | 7.6 | 7.3 | 6.9 | 6.6 |
| | SNR | 1.20 | 1.49 | 1.64 | 1.77 | 1.84 | 1.91 | 1.95 | 2.00 | 2.03 | 2.07 |
| Integrated fit | 1-$\sigma$ range [%] | 56.6 | 42.8 | 36.6 | 31.2 | 27.5 | 24.2 | 18.8 | 17.4 | 15.2 | 12.9 |
| | 2-$\sigma$ range [%] | 24.9 | 11.7 | 7.7 | 5.0 | 3.4 | 2.5 | 1.4 | 1.1 | 0.7 | 0.5 |
| | MAPE [%] | 15.8 | 11.4 | 10.0 | 9.0 | 8.3 | 7.7 | 6.8 | 6.5 | 6.3 | 5.9 |
| | MSNR | 1.57 | 1.81 | 1.93 | 2.03 | 2.06 | 2.10 | 2.13 | 2.16 | 2.18 | 2.20 |

| Number of profile-a-month | | 11 | 12 | 13 | 14 | 15 | 16 | 17 | 18 | 19 | 20 |
|---|---|---|---|---|---|---|---|---|---|---|---|
| Separated fit | 1-$\sigma$ range [%] | 15.8 | 12.8 | 13.2 | 10.4 | 9.6 | 8.5 | 7.1 | 6.7 | 6.5 | 5.8 |
| | 2-$\sigma$ range [%] | 0.7 | 0.4 | 0.4 | 0.2 | 0.2 | 0.1 | 0.1 | 0.1 | 0.1 | 0.1 |
| | MAPE [%] | 6.4 | 6.0 | 5.8 | 5.6 | 5.4 | 5.3 | 5.1 | 4.9 | 4.9 | 4.7 |
| | MSNR | 2.08 | 2.11 | 2.13 | 2.15 | 2.14 | 2.17 | 2.17 | 2.18 | 2.19 | 2.20 |
| Integrated fit | 1-$\sigma$ range [%] | 12.5 | 9.3 | 8.8 | 7.5 | 7.3 | 6.1 | 5.2 | 4.9 | 4.6 | 3.8 |
| | 2-$\sigma$ range [%] | 0.4 | 0.3 | 0.2 | 0.2 | 0.1 | 0.1 | 0.0 | 0.0 | 0.0 | 0.0 |
| | MAPE [%] | 5.7 | 5.4 | 5.2 | 5.0 | 4.9 | 4.8 | 4.5 | 4.4 | 4.4 | 4.3 |
| | MSNR | 2.20 | 2.23 | 2.24 | 2.26 | 2.24 | 2.27 | 2.26 | 2.27 | 2.27 | 2.28 |

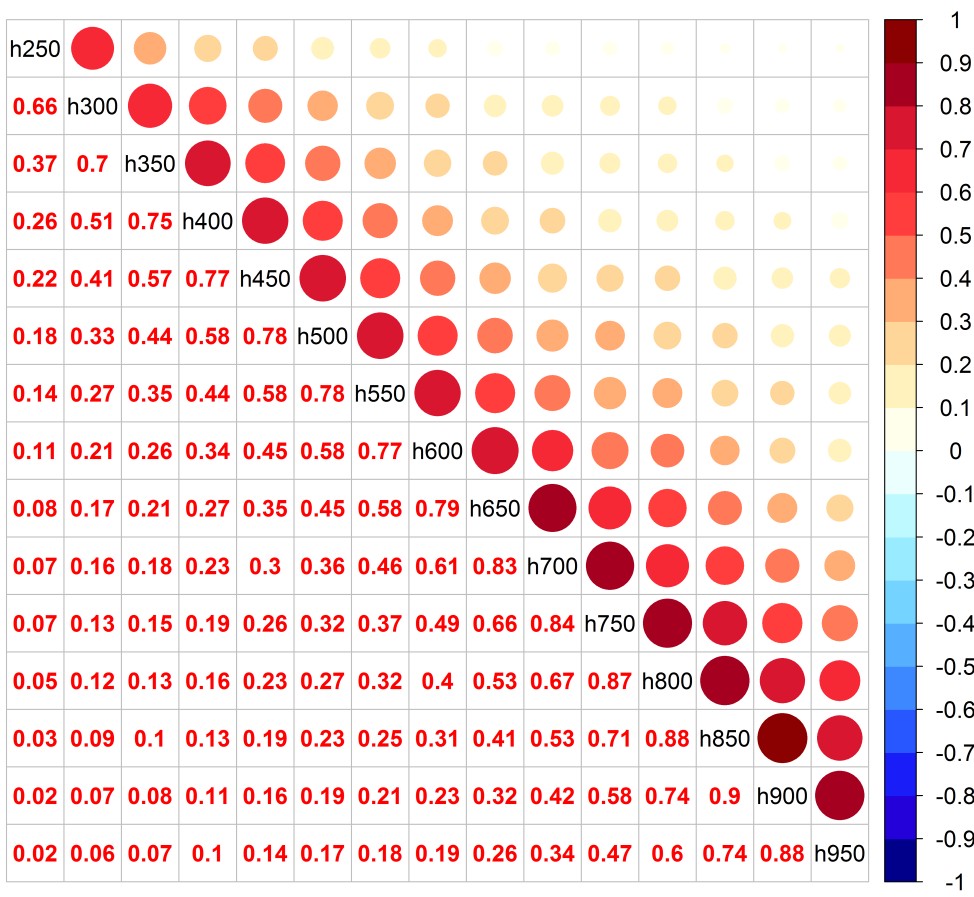

**Figure 1.** Correlation coefficients between different pressure layers, based on IAGOS profiles above western Europe.

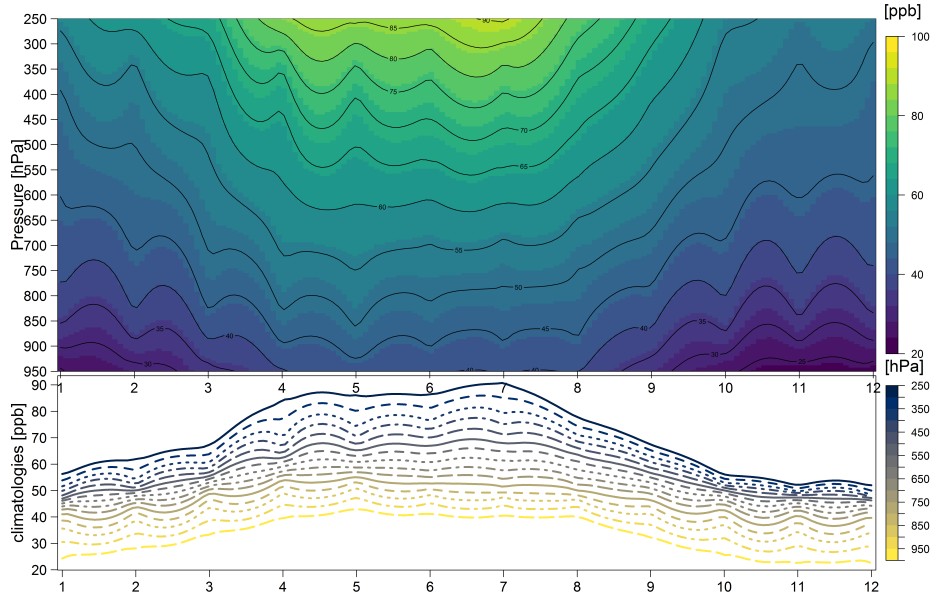

(a) Seasonal component

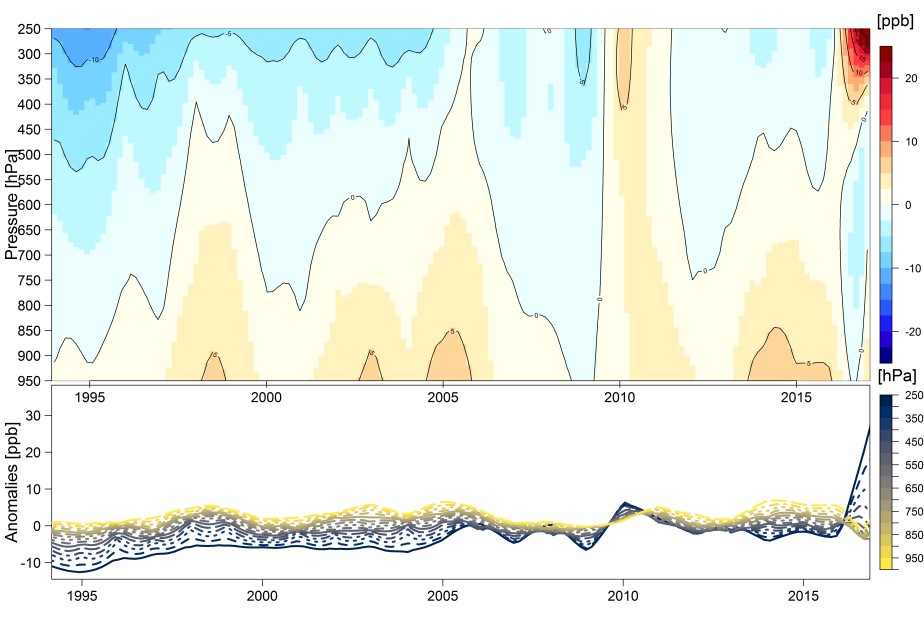

(b) Interannual component

**Figure 2.** Seasonal and interannual components for the ozone distribution above Western Europe.

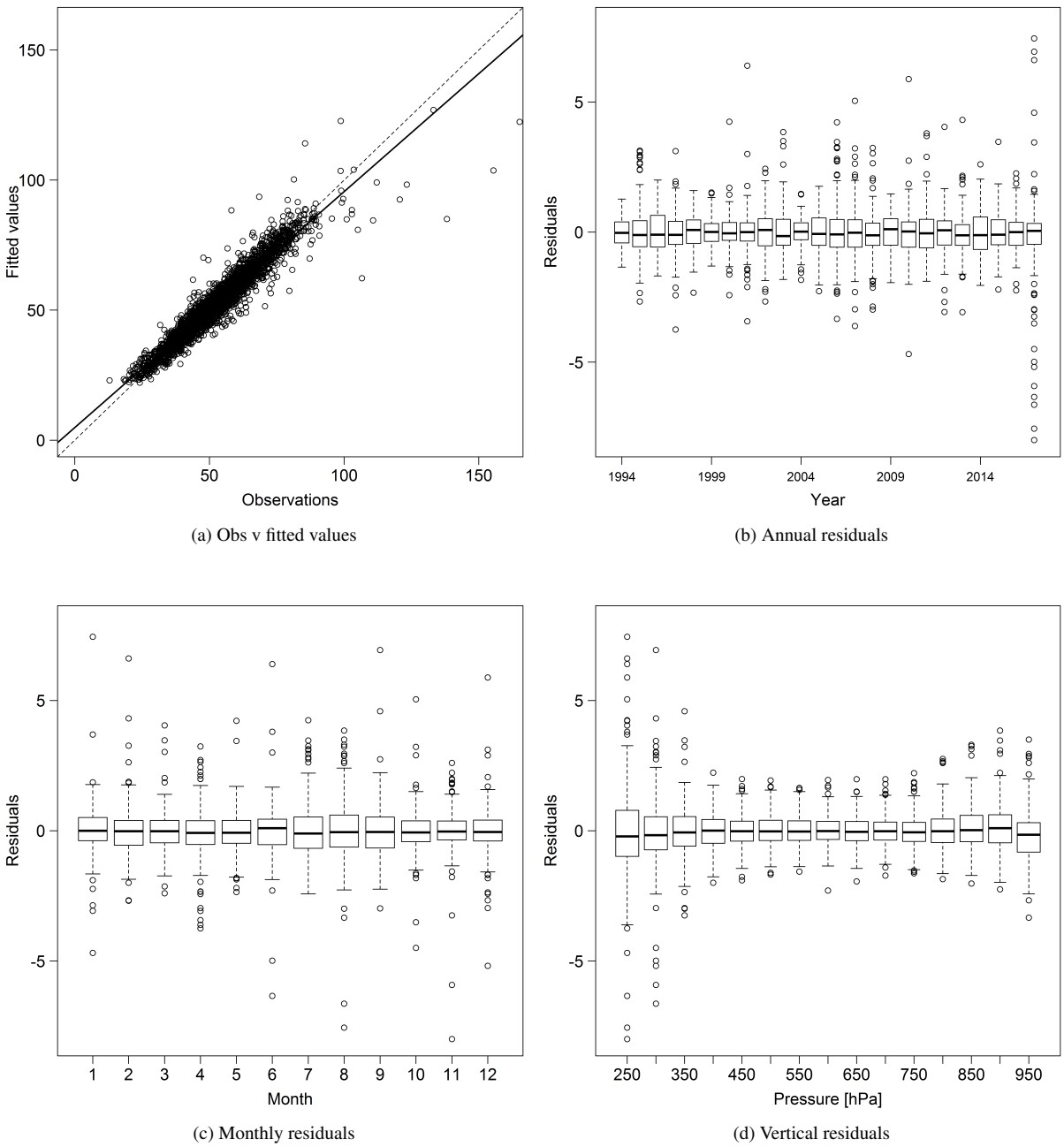

**Figure 3.** Diagnosis of statistical model fitting of the tropospheric ozone distribution above Western Europe.

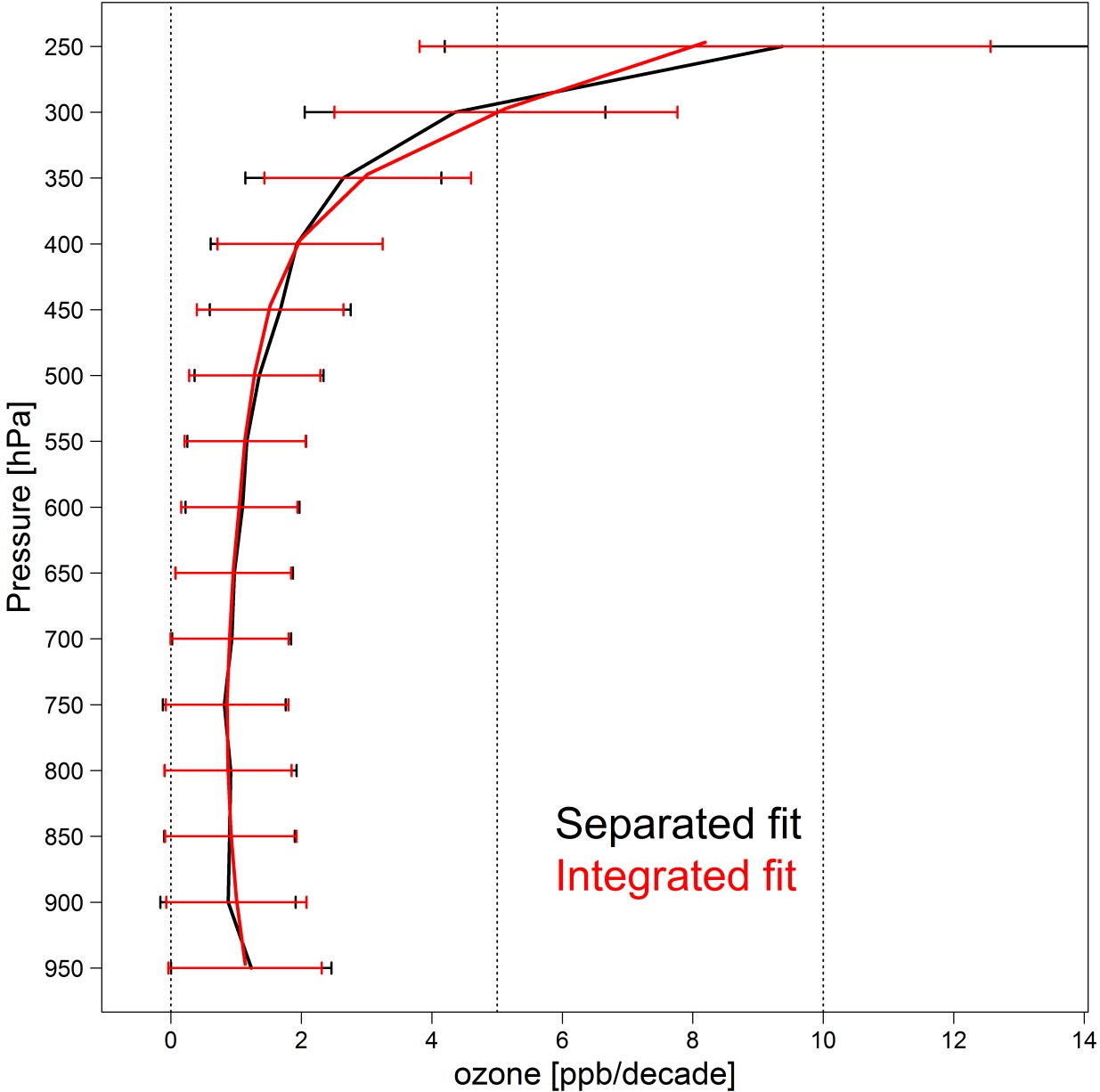

**Figure 4.** Ozone trend estimates and associated 2-sigma variabilities at 50 hPa vertical resolution above Western Europe, based on the separated fit (black) and the integrated fit (red).

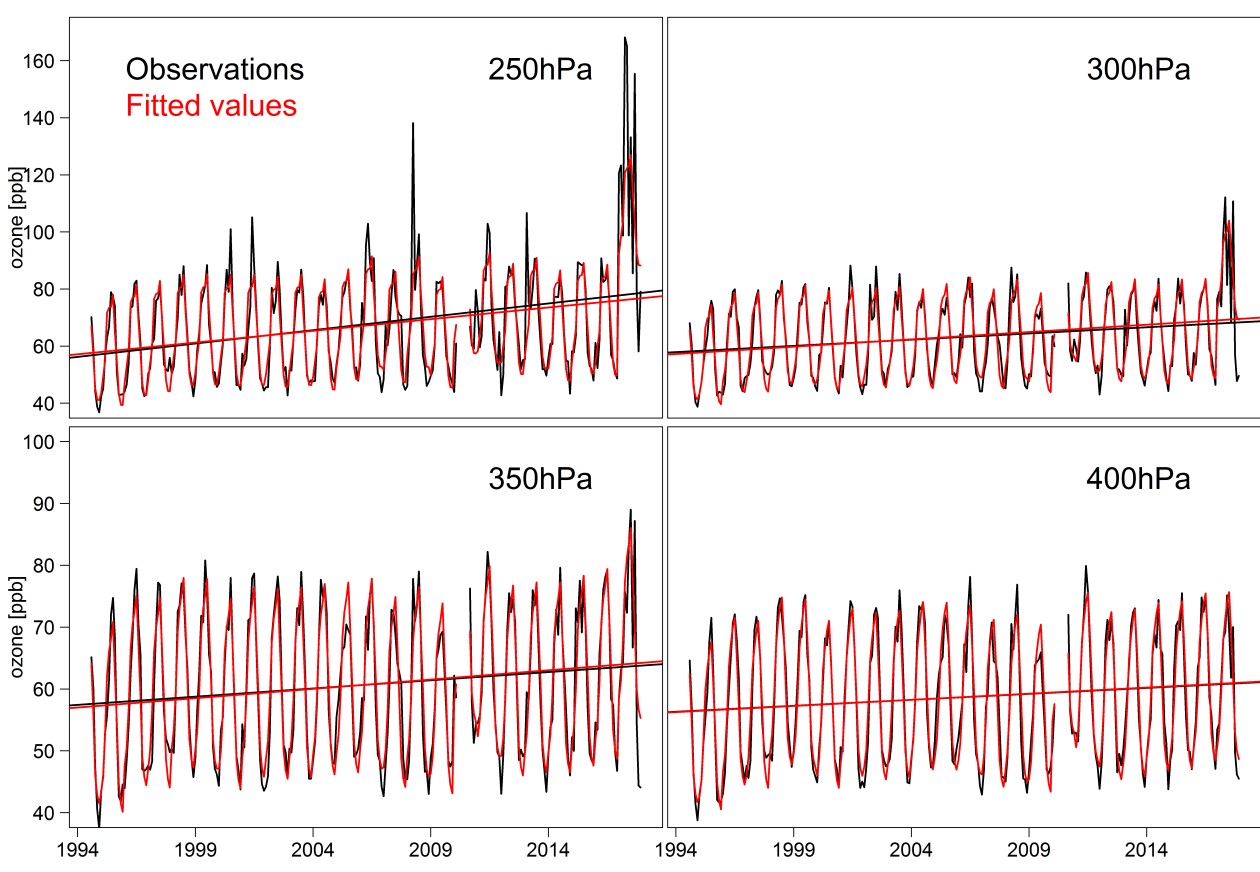

**Figure 5.** Monthly mean ozone time series and model fitted values for 4 different layers above Western Europe.

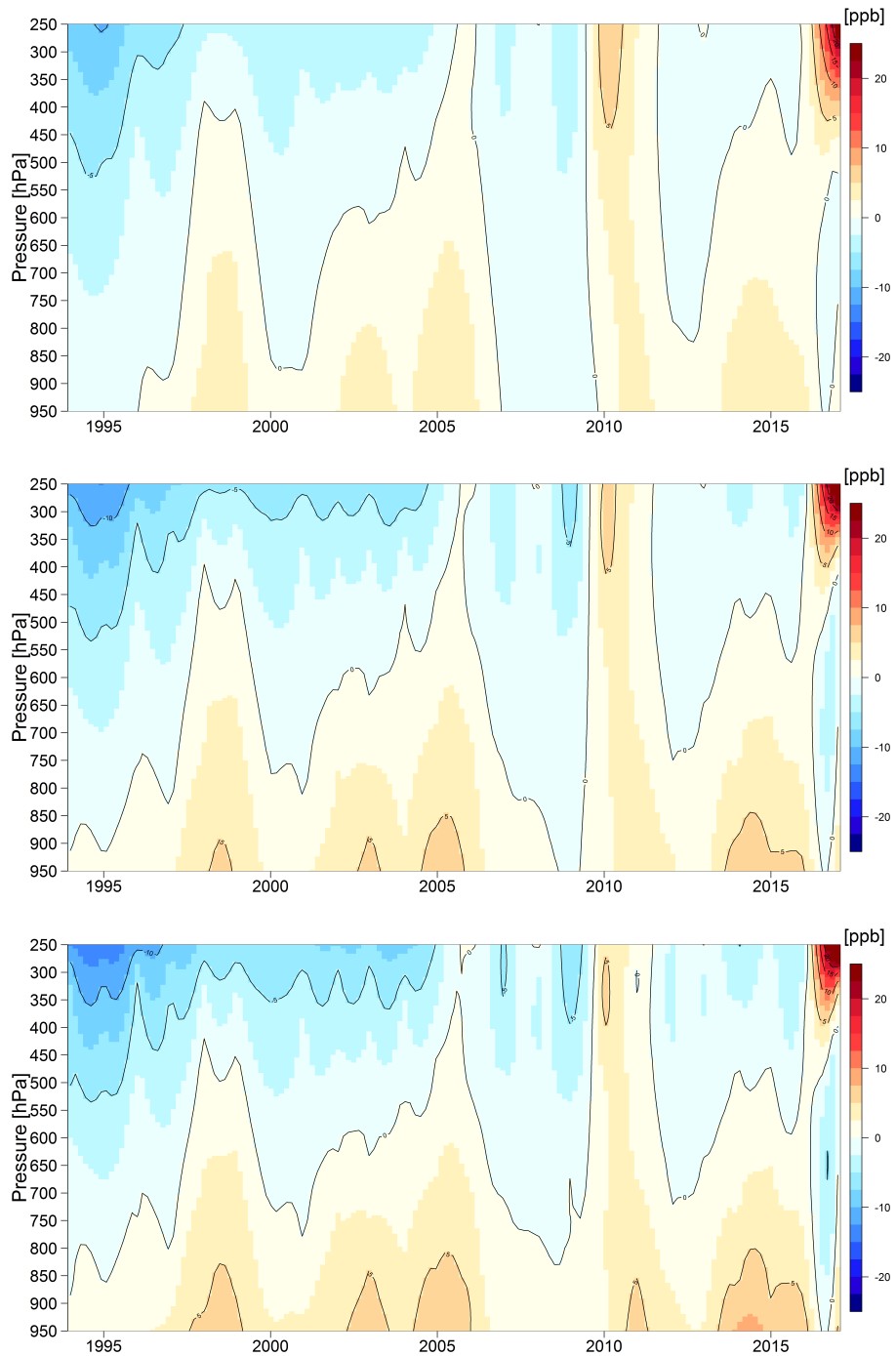

**Figure 6.** Sensitivity analysis of the fitted result from different smoothing penalties: (middle) the smoothness determined by the generalized cross validation (GCV) criterion, and potentially (upper) underfitting and (lower) overfitting by scaling the penalty coefficient ($\lambda_2$ in Equation (A2)) by a factor of 10 and 0.1, respectively.

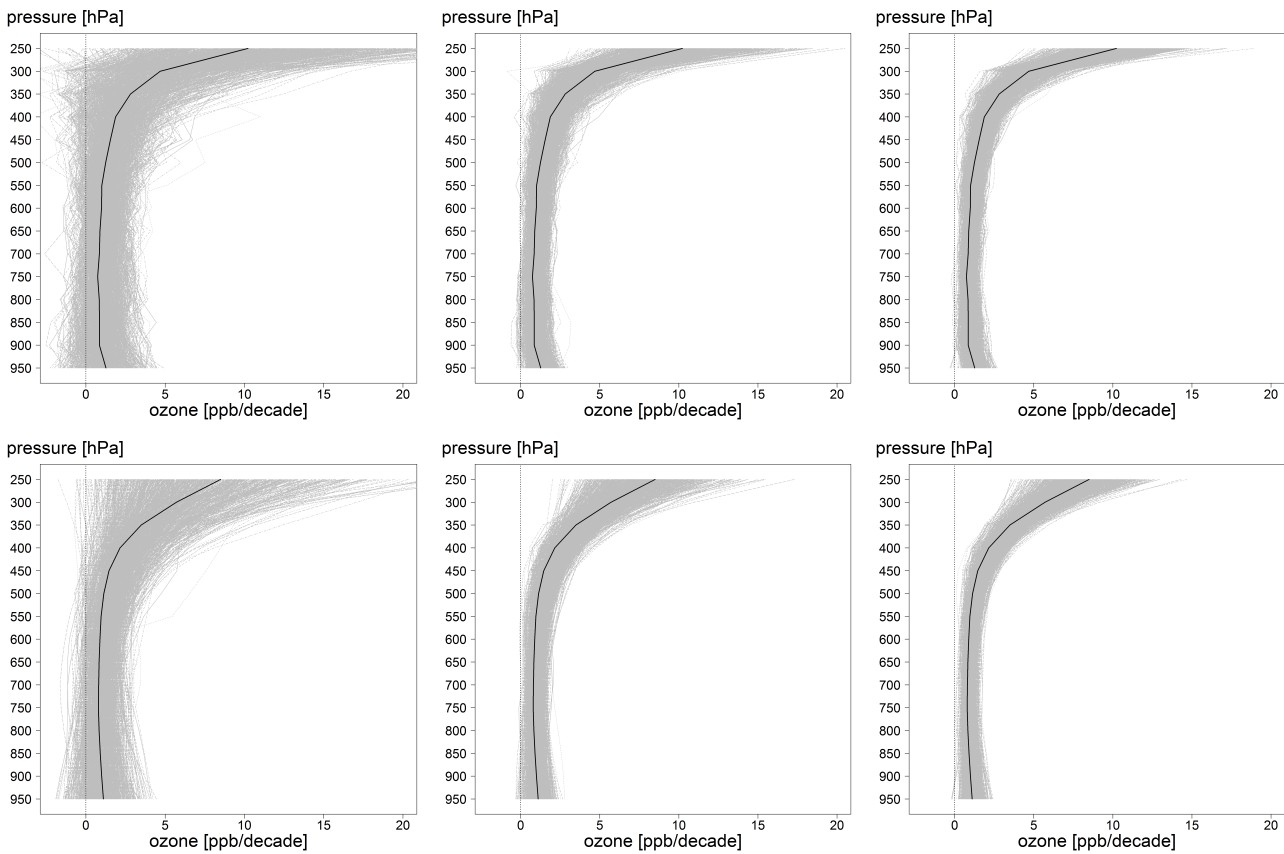

**Figure 7.** Sensitivity analysis for 1 (left), 5 (middle) and 9 (right) profiles per month based on 1000 random samples for each of 15 vertical layers above Western Europe. The analysis was conducted using the the separated fit (top) and the integrated fit (bottom). Black curves represent the vertical distribution of the true trends, based on the full IAGOS data set.

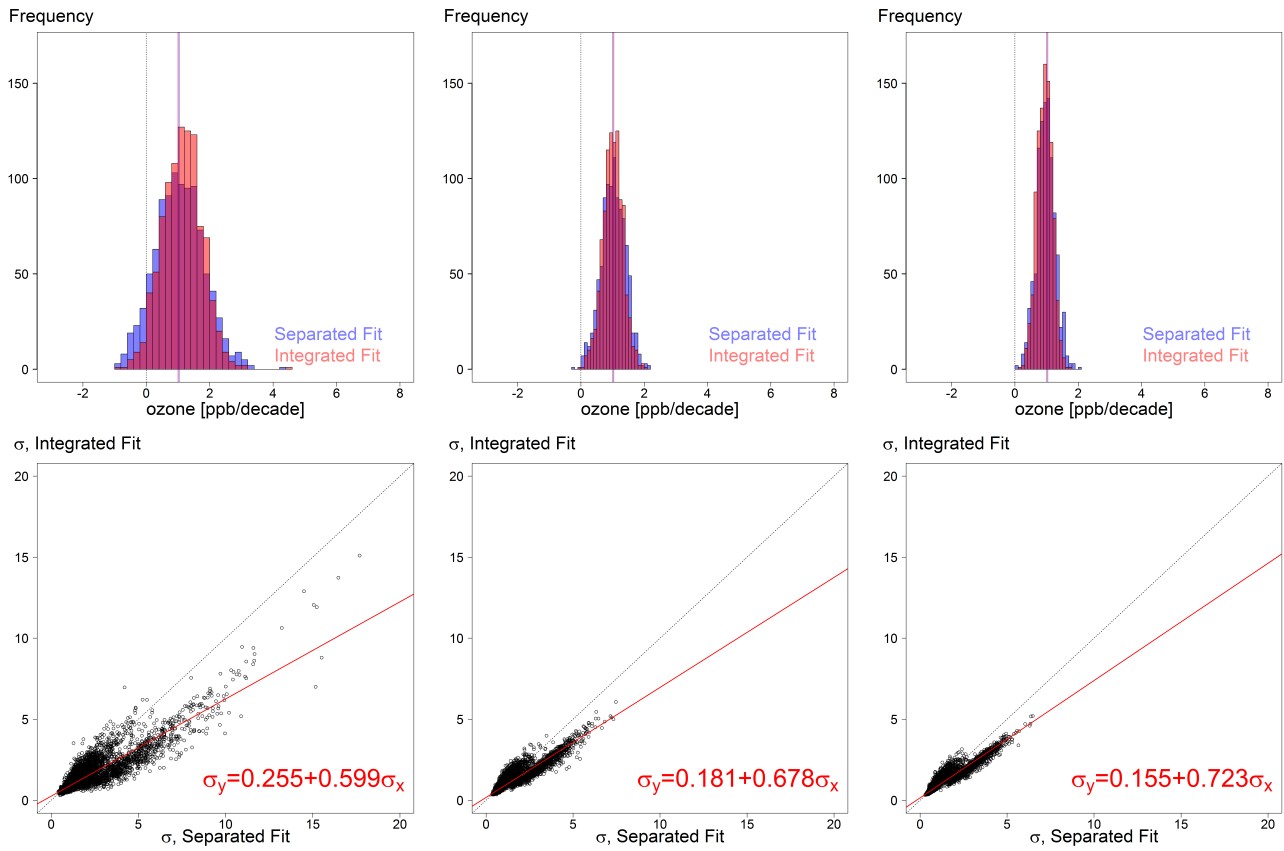

**Figure 8.** Sensitivity analysis at 550 hPa based on 1000 random samples of 1 (left), 5 (middle) and 9 (right) selected profiles per month. Shown are the sampled trend distributions with vertical lines indicating the trend values derived from the full data set (top), and the corresponding matched trend uncertainties based on the separated and integrated fits (bottom).

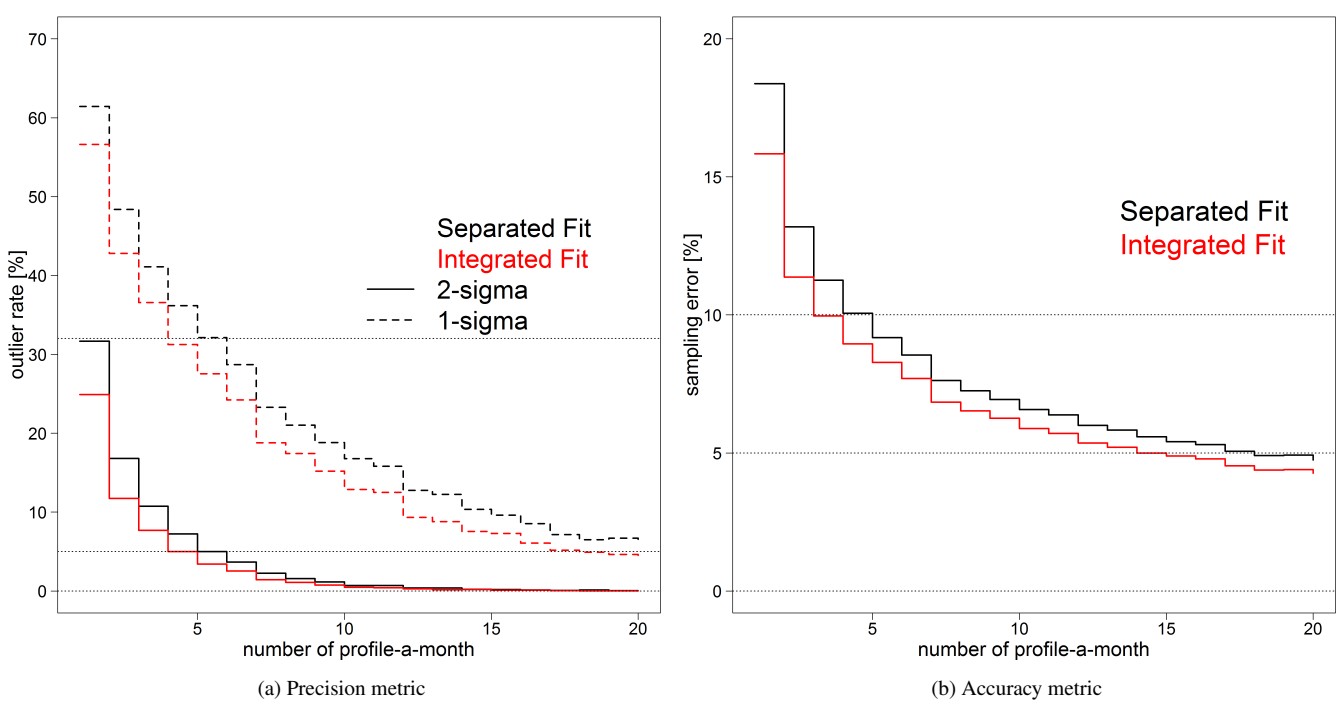

(a) Precision metric         (b) Accuracy metric

**Figure 9.** The marginal decrement of the outlier rate (sampled trends located outside 1- or 2-$\sigma$ interval of the assumed true trend) (left) and the sampling error (mean absolute percentage error) (right).

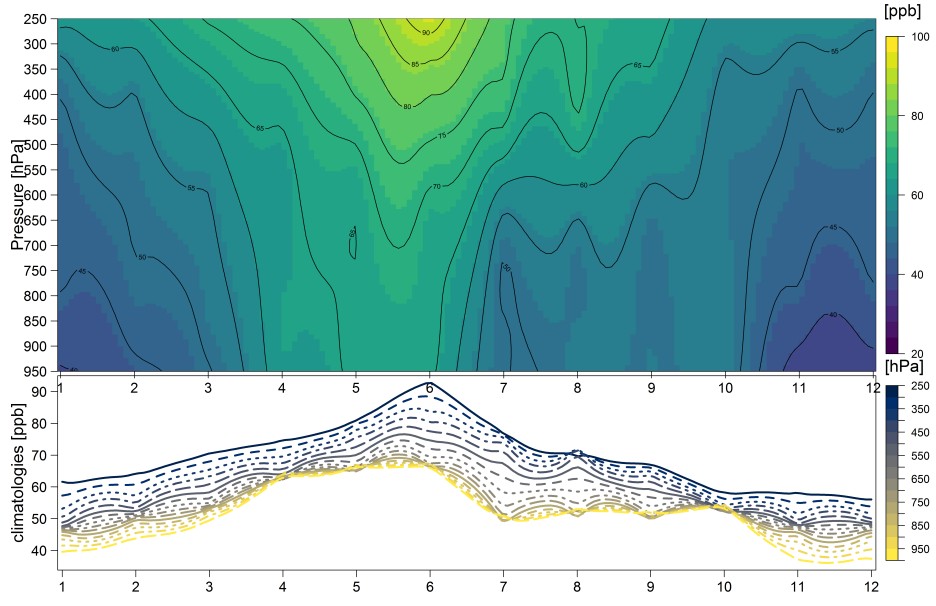

(a) Seasonal component

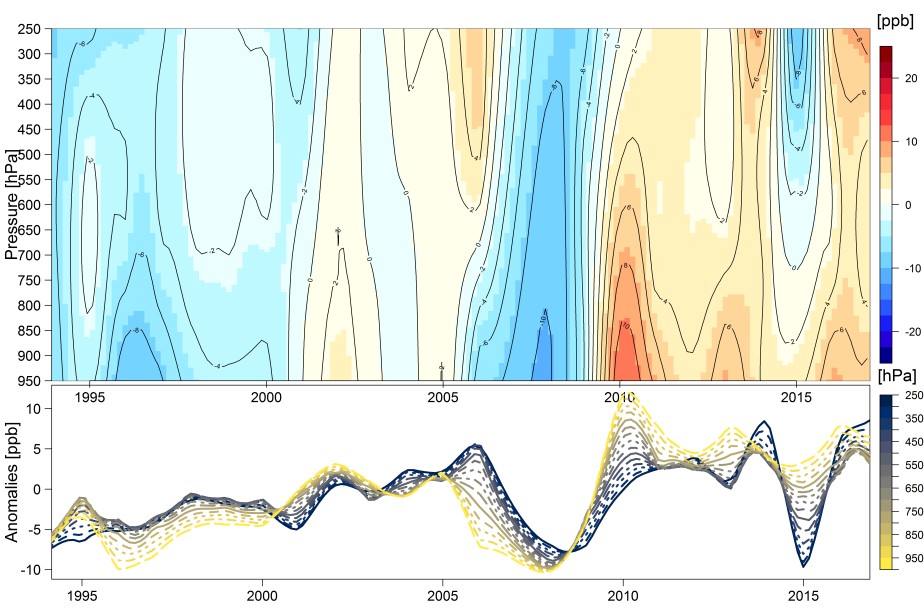

(b) Interannual component

**Figure 10.** Seasonal and interannual components for the ozone distribution above NE China.

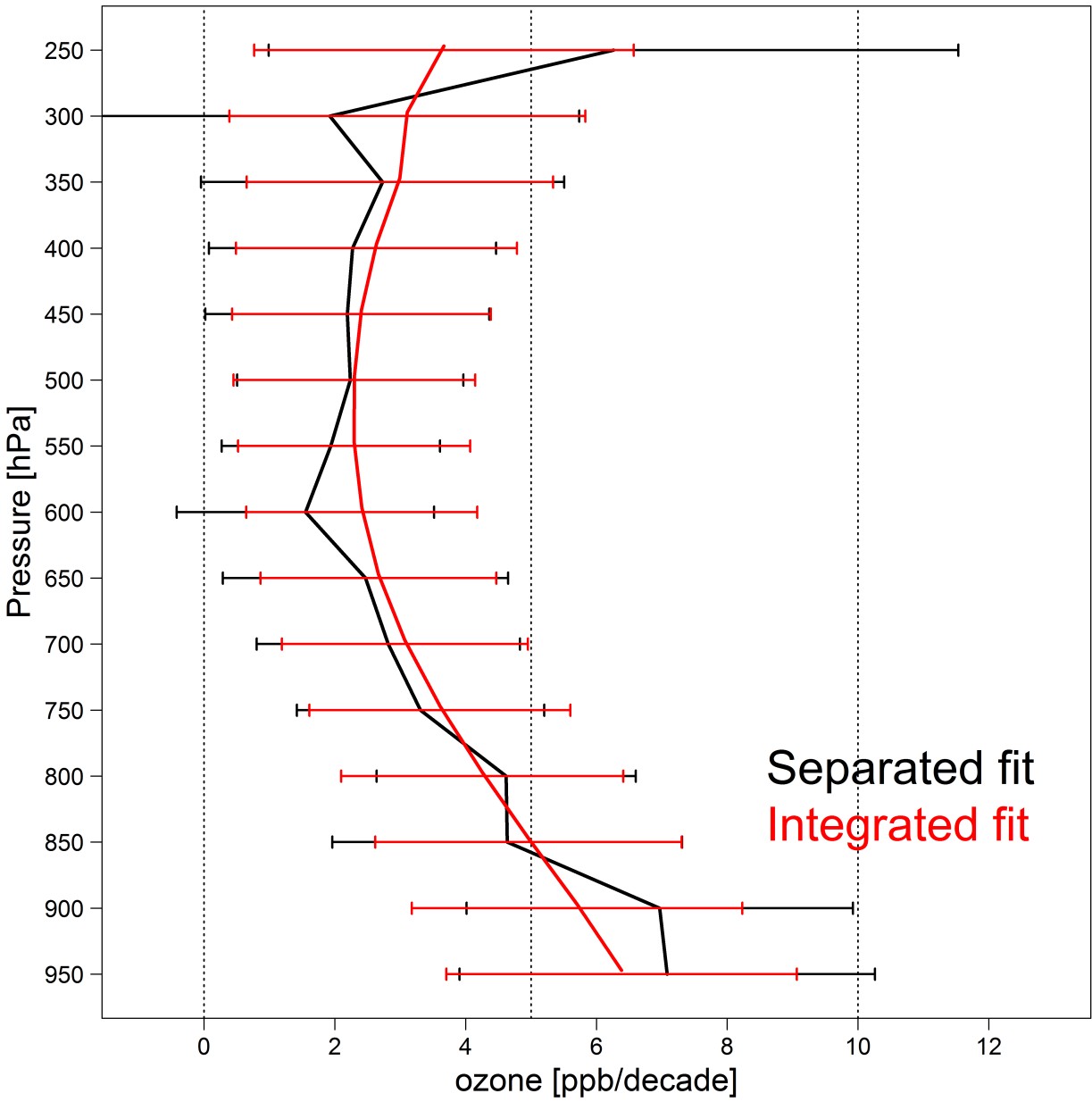

**Figure 11.** Ozone trend estimates and associated 2-sigma variabilities at 50 hPa vertical resolution above NE China, based on the separated fit (black) and the integrated fit (red).

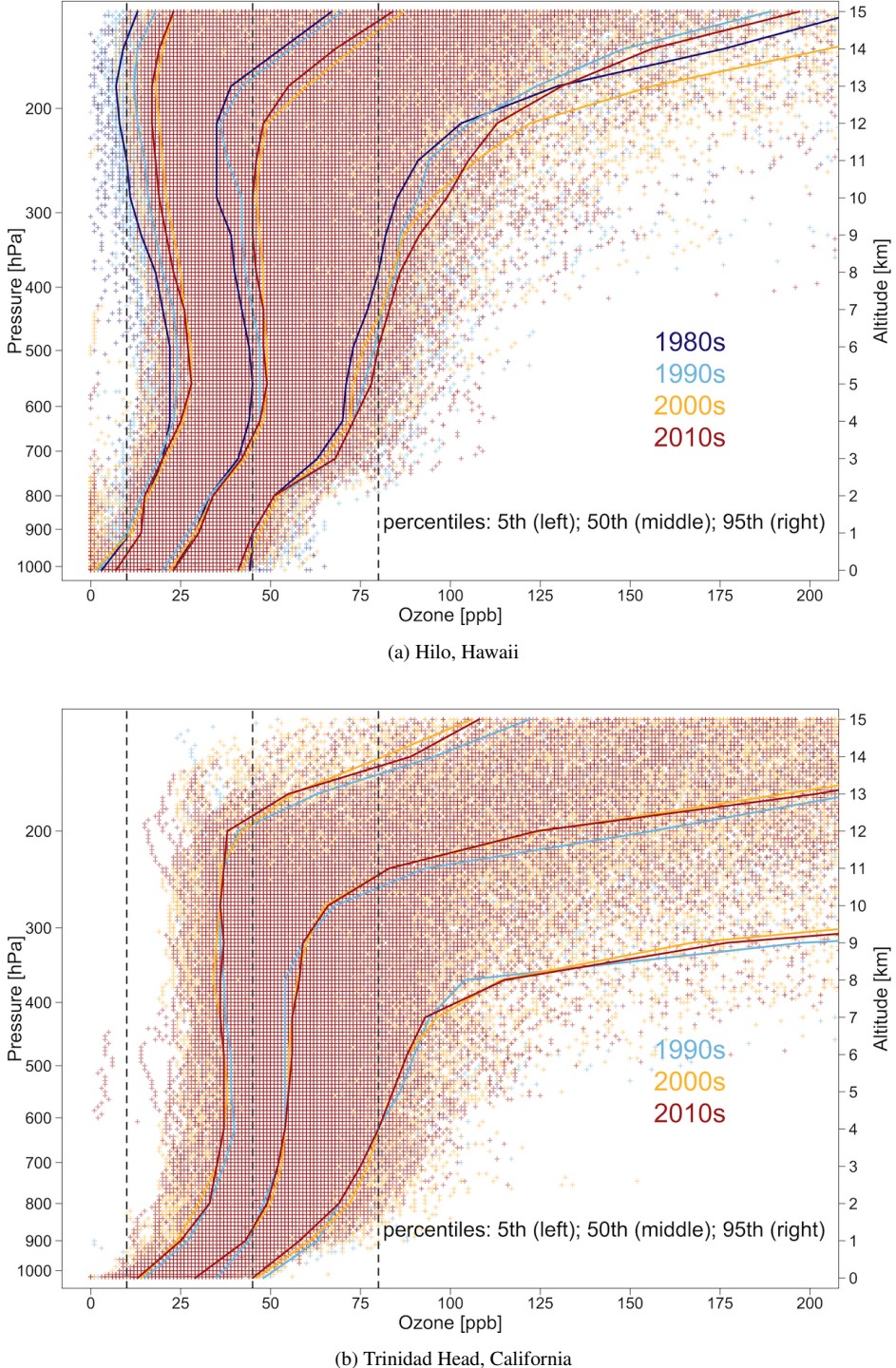

(a) Hilo, Hawaii

(b) Trinidad Head, California

**Figure 12.** All ozone observations below 15 km as measured by ozonesondes above Hilo, Hawaii and Trinidad Head, California. The observations are colored according to the decade in which they were measured. The solid lines represent the 5th, 50th and 95th percentiles. The vertical lines represent a reference at 10, 40 and 80 ppb.

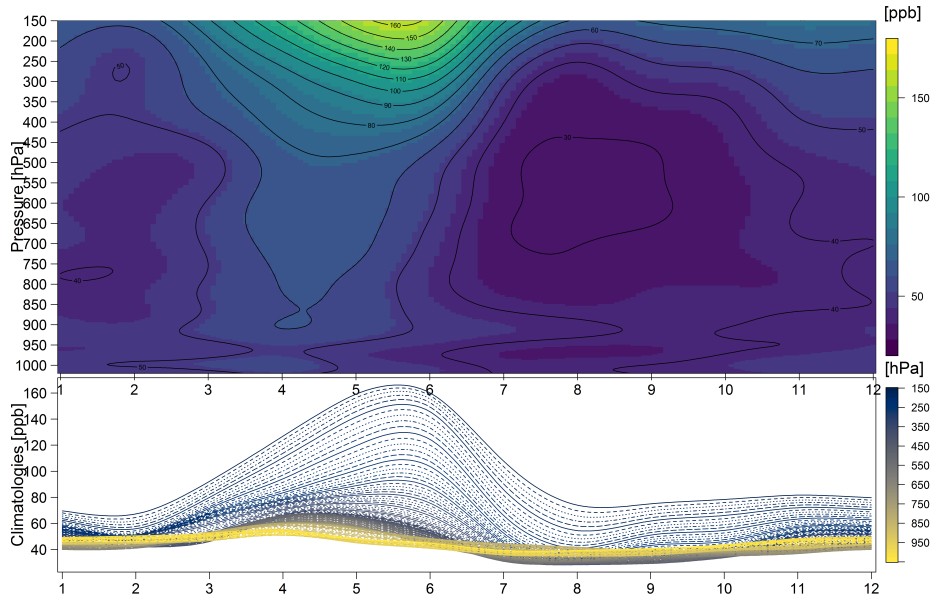

(a) Seasonal component

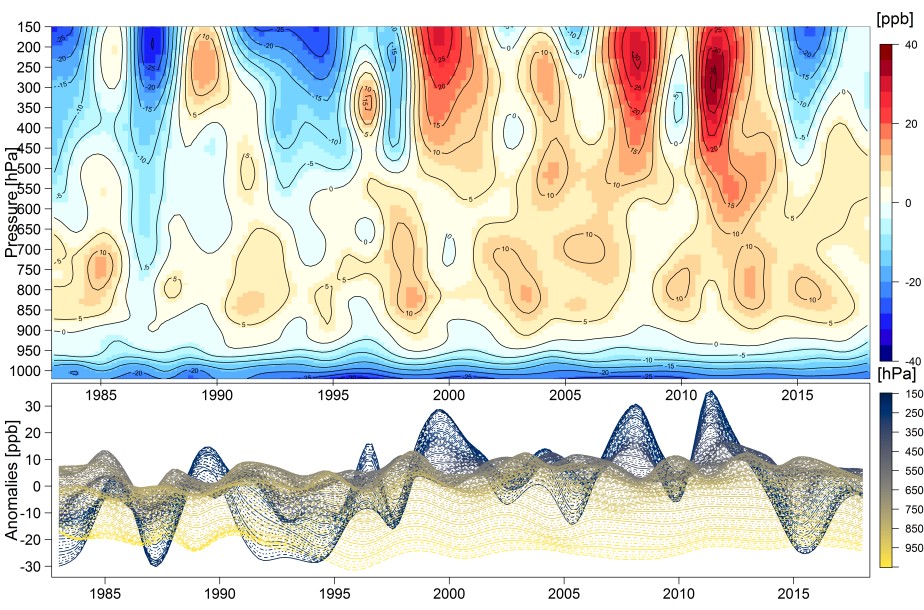

(b) Interannual component

**Figure 13.** Seasonal and interannual components for the ozone distribution above Hilo, Hawaii.

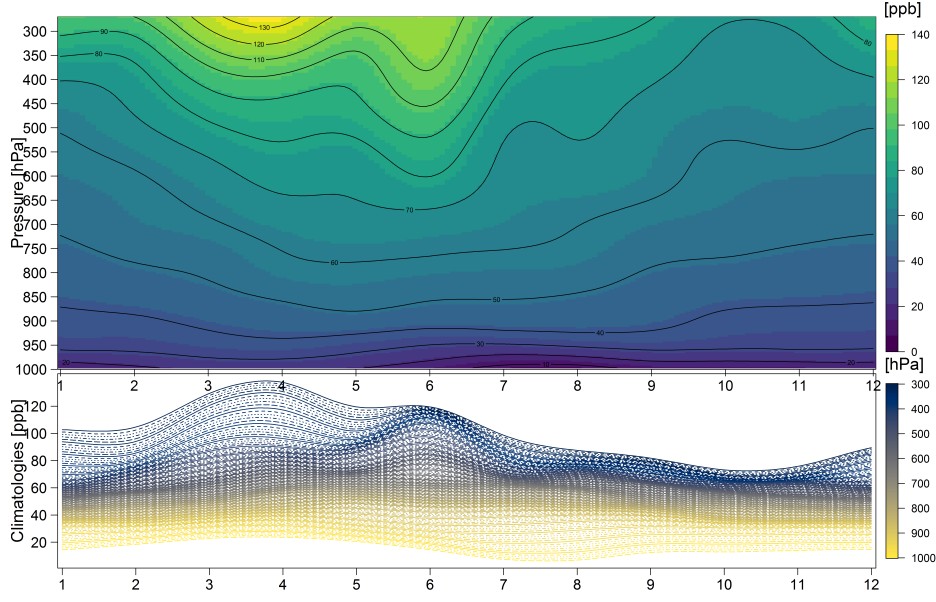

(a) Seasonal component

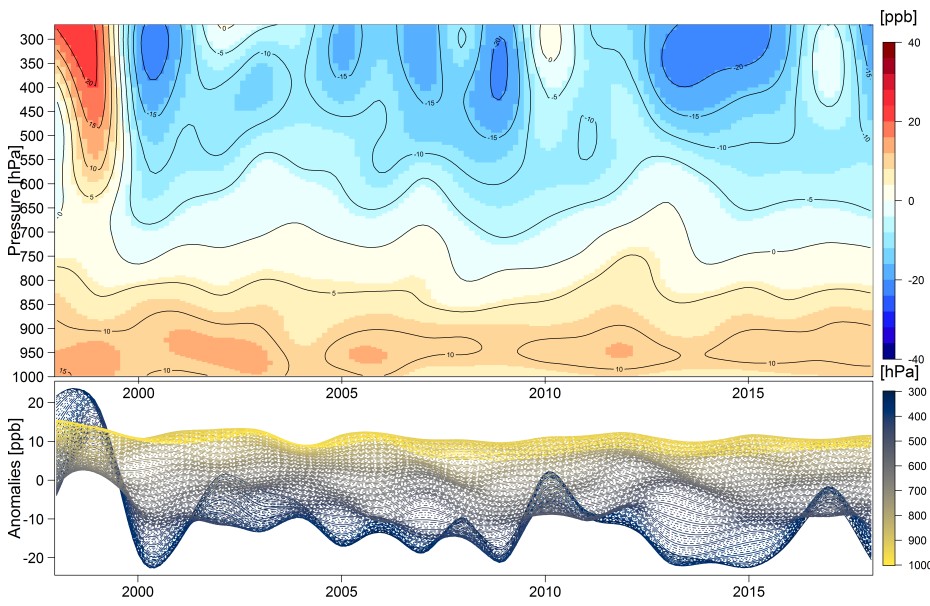

(b) Interannual component

**Figure 14.** Seasonal and interannual components for the ozone distribution above Trinidad Head, California.

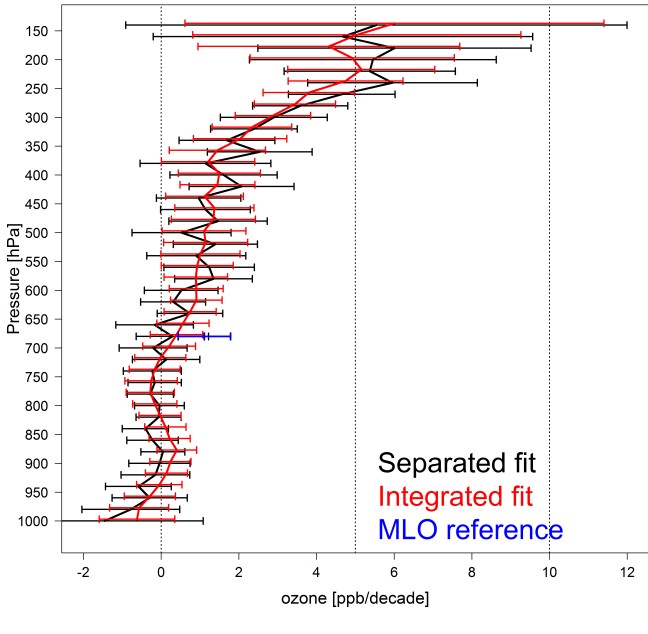

(a) Hilo, Hawaii (1982-2018)

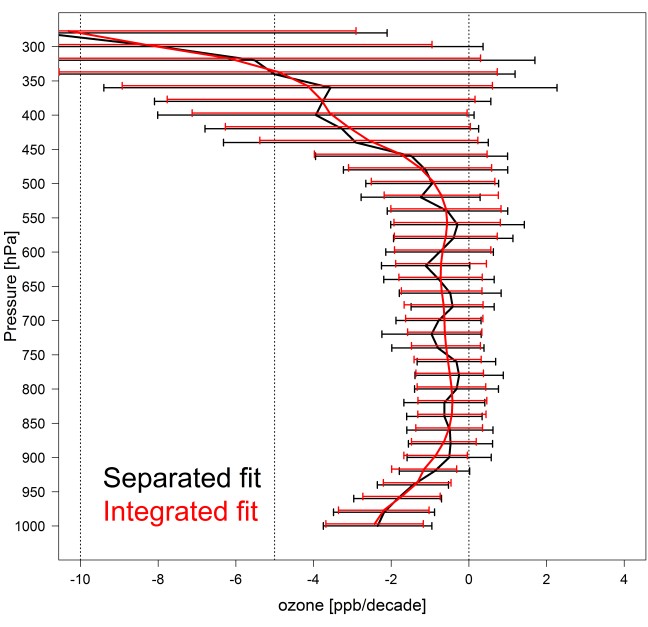

(b) Trinidad Head, California (1998-2018)

**Figure 15.** Ozone trend estimates and associated 2-sigma variabilities at 20 hPa vertical resolution, based on the separated fit and integrated fit methods above Hilo, Hawaii (with the ozone trend at Mauna Loa Observatory at 680 hPa provided for reference (1982-2018)) and Trinidad Head, California.