# Peer review of "Statistical regularization for trend detection: An integrated approach for detecting long-term trends from sparse tropospheric ozone profiles"

_Atmospheric Chemistry and Physics, 2019_

## Referee Comment (RC1) · Anonymous Referee #2 · 1 Jun 2020

My initial impression and doubts remain. The paper is well written and presented, the discussion accurate although quite technical. Overall scopes are, in my opinion, quite narrow. I still see the manuscript better suited for a journal with a more speciliased focus on the subject of statistical methods, or on measurements.

Anyhow, the subject covered here is explicitely mentioned in the ACP subject areas (https://www.atmospheric-chemistry-and-physics.net/about/subject_areas.html) and several references used by the authors confirm that other studies dealing with trend detection and tropospheric measurements have been published in ACP.

[Figure]

I therefore advice the editor to accept the manuscript for publication, and suggest the authors to simplify the discussion where possible, to reach out larger audience. Perhaps some technical material can be moved to appendix?

Moreover, what about comparing the result of your methodology with profiles treated with other methods mentioned by the authors in the introduction? this would help determine what is the cost-effectiveness with respect to benefits of your approach compared to simpler ones.
* * *

---

## Referee Comment (RC2) · Anonymous Referee #1 · 4 Jun 2020

The article proposes a new method to compute temporal trends in ozone vertical profiles. They take stock of auto-correlation in the vertical dimension to minimize uncertainties in the estimated trends.

Most of the paper is devoted to statistical considerations, with very little geophysical discussions, so that I agree with Anonymous Referee #2 that such a paper would have been more suited for a statistical journal.

But the manuscript remains well written and the topic of trend detection for tropospheric

ozone is important and relevant to Atmos. Chem. Phys. so that I see little ground for not supporting its publication.

General comment

My main frustration is that the added value of the integrative method is only tested in terms of trend estimate, whereas in atmospheric ozone studies the significance of the trend is also often challenging to assess. The benefit of this method in terms of significance would deserve to be discussed.

It is not fully clear how European and Chinese IAGOS profiles are treated but an "aggregate" is mentioned P8L1. It is very questionable to average all profiles collected over a whole continent, and subsequently seek to assess such subtilities in the estimated trend.

The benefit of the new method could be considered relatively marginal. It would be interesting to provide more discussion on tropospheric ozone trends to demonstrate the importance to refine these estimates.

Specific comments

In several occasions in the manuscript, I would avoid the term "noise". The vertical structures documented by atmospheric profiles are not due to instrument random uncertainty but they carry an actual "signal" to understand atmospheric variability. I understand that this variability is not in the focus for long term trend studies, but they can not be considered as "noise". In turn, the regularization applied in eq 4 is very questionable, and the impact of this step on the overall findings should be assessed through a sensitivity study.

Abstract: Add the number of years available in GMD/IAGOS

P8L30: a geophysical interpretation of the 2010 anomaly should be provided.

P8L32: the figure does not provide anomaly for the seasonal cycle, so that this statement is poorly supported

P10L18: I find figure S4 more insightful than Figure 6, suggest swapping both

P11L22: which geophysical process could explain the difference vertical correlation depending on the region?

P12L2: add percentiles in the figure legend

P14L5: this sentence seems misplaced

---

## Author Comment (AC1) · 8 Jul 2020

Please see the attached pdf file for the response to both reviewers.

Please also note the supplement to this comment:
https://www.atmos-chem-phys-discuss.net/acp-2019-959/acp-2019-959-AC1-supplement.pdf

2020.

---

## Author Response (AR1)

We thank the reviewer for providing valuable comments on our manuscript. The reviewer comments are shown below in bold font, followed by our response in normal font.

**The article proposes a new method to compute temporal trends in ozone vertical profiles. They take stock of auto-correlation in the vertical dimension to minimize uncertainties in the estimated trends.**

**Most of the paper is devoted to statistical considerations, with very little geophysical discussions, so that I agree with Anonymous Referee #2 that such a paper would have been more suited for a statistical journal. But the manuscript remains well written and the topic of trend detection for tropospheric ozone is important and relevant to Atmos. Chem. Phys. so that I see little ground for not supporting its publication.**

**My main frustration is that the added value of the integrative method is only tested in terms of trend estimate, whereas in atmospheric ozone studies the significance of the trend is also often challenging to assess. The benefit of this method in terms of significance would deserve to be discussed.**

Thanks for pointing out the evaluation of method in terms of the significance, we added an additional analysis and discussion in Section 3.2 as follows:

*The above discussion is focused on the precision and accuracy of the trend estimate at various sampling frequencies. In addition, we can explore the impact of sampling frequency on our ability to simply detect the presence of a trend, based on uncertainty analysis. In order to evaluate the resulting uncertainty associated with the sampled trends, we include the mean signal-to-noise ratio (MSNR) between trends derived from the full data set (the assumed true trend values) and the uncertainty in each sample (i.e. standard error associated with the trend estimate) in Table 1. In the interest of a fair comparison, this calculation is done by comparing the sampling uncertainty with trend values derived by the same method. Note that we do not use the concept of ``statistical significance'' to indicate evidence for the trends, following the recent recommendations from the American Statistical Association (Wasserstein and Lazar, 2016; Wasserstein et al., 2019). Instead, the benchmark we selected for comparing trend uncertainties is a rejection of the null hypothesis at the 95\% confidence when the SNR exceeds a threshold value of 2. As noted in Table 1 we already knew the MSNR is around 2.3 for both methods when using the full datasets, however, the SNR analysis shows that the benchmark value of 2 can be achieved at a sampling frequency of just four profiles per month when using the integrated fit, whereas the separated fit requires 8 profiles per month. In summary, at a low sampling frequency the integrated fit provides not only more precise and accurate trend estimates, but the uncertainty associated with trends can be reduced.*

Wasserstein, RL and Lazar, NA. 2016. The ASA's Statement on p-Values: Context, Process, and Purpose. The American Statistician 70: 129–133. DOI: 10.1080/00031305.2016.1154108

Wasserstein, RL, Schirm, AL and Lazar, NA. 2019. Moving to a World Beyond "p < 0.05". The American Statistician 73(sup1): 1–19. DOI: 10.1080/00031305.2019.1583913

**It is not fully clear how European and Chinese IAGOS profiles are treated but an "aggregate" is mentioned P8L1. It is very questionable to average all profiles collected over a whole continent, and subsequently seek to assess such subtilities in the estimated trend.**

The IAGOS datasets in this study are from limited regions and are not aggregated across the entire continent of Europe or across the entire country of China.  We now state in Section 2.2 (where the IAGOS data are introduced) that the Europe and China datasets focus on limited regions of central-Western Europe (0-15 E, 47-55 N) and northeast China/South Korea (110-135 E, 28-45 N).

We added a review of IAGOS observations to demonstrate the regional representativeness of IAGOS data in the beginning of Section 3:

*Several studies have demonstrated that IAGOS data above Western Europe are consistent with ozonesonde records in the UTLS (Staufer et al., 2013, 2014), and the data have compared well to regional surface and free tropospheric ozonesonde records (Thouret et al., 1998; Logan et al., 2012; Petetin et al., 2018). Petetin et al. (2018) validated IAGOS data in the lowest troposphere over Western Europe against rural monitoring sites at various elevations and concluded that those observations can be used to study the air quality in the agglomeration. Nevertheless, to reduce the influence from the strong diurnal cycle at the surface, all observations below the 950 hPa level and within 300 m of the surface were removed from the analysis (Gaudel et al., 2020).*

Gaudel, A., Cooper, O. R., Chang, K.-L., Bourgeois, I., Ziemke, J. R., Strode, S. A., Omen, L. D., Sellitto, P., Granier, C., Nédélec, P., Blot, R., and Thouret, V.: Tropospheric ozone is still increasing across the Northern Hemisphere, Sci. Adv. (submitted), 2020.

Logan, J. A., Staehelin, J., Megretskaia, I. A., Cammas, J. P., Thouret, V., Claude, H., et al. (2012). Changes in ozone over Europe: Analysis of ozone measurements from sondes, regular aircraft (MOZAIC) and alpine surface sites. Journal of Geophysical Research: Atmospheres, 117(D9).

Petetin, H., Jeoffrion, M., Sauvage, B., Athier, G., Blot, R., Boulanger, D., et al. (2018). Representativeness of the IAGOS airborne measurements in the lower troposphere. Elem Sci Anth, 6(1).

Staufer, J., Staehelin, J., Stubi, R., Peter, T., Tummon, F., & Thouret, V. (2013). Trajectory matching of ozonesondes and MOZAIC measurements in the UTLS-Part 1: Method description and application at Payerne, Switzerland. Atmospheric Measurement Techniques, 6(12), 3393-3406.

Staufer, J., Staehelin, J., Stubi, R., Peter, T., Tummon, F., & Thouret, V. (2014). Trajectory matching of ozonesondes and MOZAIC measurements in the UTLS: Part 2. Application to the global ozonesonde network. Atmospheric Measurement Techniques, 7(1), 241-266.

Thouret, V., Marenco, A., Logan, J. A., Nédélec, P., & Grouhel, C. (1998). Comparisons of ozone measurements from the MOZAIC airborne program and the ozone sounding network at eight locations. Journal of Geophysical Research: Atmospheres, 103(D19), 25695-25720.

**The benefit of the new method could be considered relatively marginal. It would be interesting to provide more discussion on tropospheric ozone trends to demonstrate the importance to refine these estimates.**

In addition to the above new demonstration and discussion, we also provided a new discussion in the Conclusion as follows:

*Detecting trends of tropospheric ozone from ozonesonde profiles is challenging due to relatively low sampling frequency combined with high temporal variability. Regularization is a statistical learning tool which makes a trade-off between, 1) high fitted bias (low flexibility) that results from underfitting, and 2) high sensitivity to small data fluctuations (low generalizability) that results from overfitting. The underfitting can be avoided by making sure the number of basis functions (e.g. thin-plate splines) are large enough to represent the underlying process. A model can be considered to be overfitted if a high cross validation error is found (which can be investigated by, e.g., iteratively removing one observation and predicting this value from the remaining observations). In terms of detecting tropospheric ozone trends from vertical profiles, the vertical correlated structures in the neighboring pressure layers can be used to inform the learning process. The benefit of this approach can be reflected by the detectable trends (if any) at a low sampling rate (i.e. we have higher confidence of our ability to detect a trend from weekly sampled ozonesonde data), and by an improved quantification of trend estimates in terms of accuracy and precision.*

*This technique efficiently reduces the uncertainty of the resulting trends, and thus increases our ability to detect and quantify trends of smaller magnitude. Therefore, this method is valuable for improved trend detection of ozonesonde records, because although these records are sufficiently long-term and have high vertical resolution with high accuracy, standard trend analysis is still challenging due to the  limited sampling rate. This refined estimation is also expected to be beneficial when comparing trends derived from different regions or observing systems, since the result is less sensitive to incoherent or unstructured variations.*

**Specific comments**

**In several occasions in the manuscript, I would avoid the term "noise". The vertical structures documented by atmospheric profiles are not due to instrument random uncertainty but they carry an actual "signal" to understand atmospheric variability. I understand that this variability is not in the focus for long term trend studies, but they can not be considered as "noise". In turn, the regularization applied in eq 4 is very**

**questionable, and the impact of this step on the overall findings should be assessed through a sensitivity study.**

We replaced the term "noise" by short-lived anomalies or unstructured variation in the manuscript. The only exception was made for the discussion of "signal to noise ratio", as this is a well known relative concept. We also added a new analysis and discussion on the regularization in Section 3.1:

*To demonstrate the role of regularization in the model fitting, we first provide a synthetic example that illustrates the problems of underfitting (the model is not flexible enough to capture the general pattern in the data) and overfitting (the model is overparameterized, so some unphysical fluctuations are present) in supplementary Figure S1. Neither underfitting nor overfitting is an appropriate representation of the true process. Once sufficient model complexity is supplied (e.g. placing a knot for spline functions every 50 hPa), the statistical regularization can be used to penalize overly complex models and thus prevent overfitting. The overfitting of a surface is less obvious than a curve, but we provide a demonstration of the fitting by adjusting the optimized penalty coefficient (which is selected by the generalized cross validation (Wood, 2006) and is proven to be reliable, as discussed above). We scale the penalty coefficient ($\lambda_2$ in Equation (A2) while keeping $\lambda_1$ fixed) by a factor of 10 and 0.1, respectively (i.e. the smaller penalty, the more roughness will be present). The result is shown in Figure 6: the underfitting can be seen as an over-smoothed representation of the underlying structure, and sharp variations (e.g. overly complicated roughness) are an indication of overfitting.*

[Figure]

Figure S-1: A synthetic prole for the illustration of the issue of underfitting and overfitting. A synthetic profile for the illustration of the issue of underfitting and overfitting. Data points are generated by adding random error to the function $x = sin(y/100) \times (y/100)^2$ (black curve). The appropriate fit should closely follow the true process (red curve), while the underfit indicates that the result failed to represent the general pattern of the true process (blue curve), and the overfit indicates that the result is overly complicated and is influenced by the noise component (green curve). Note that these three fits are based on the same model specification, except that we adjust the roughness penalty to illustrate the underfit and overfit.

[Figure]

Figure 6. Sensitivity analysis of the fitted result from different smoothing penalties: (middle) the smoothness determined by the generalized cross validation (GCV) criterion, and potentially underfitting (upper) and overfitting (lower) by scaling the penalty coefficient (lambda_2 in Equation (A2)) by a factor of 10 and 0.1, respectively.

**Abstract: Add the number of years available in GMD/IAGOS**

The study period for each data set was added.

**P8L30: a geophysical interpretation of the 2010 anomaly should be provided.**

We added a clarification as following:

*The 22-year time series has one data gap spanning March-August 2010 when instrument failures resulted in just a few sporadic profiles (Petetin et al. 2016b) (the 2010 annual mean is represented by the other 6 months, after adjustments for seasonality).*

*Petetin, H., Thouret, V., Fontaine, A., Sauvage, B., Athier, G., Blot, R. et al. (2016). Characterising tropospheric O3 and CO around Frankfurt over the period 1994-2012 based on MOZAIC-IAGOS aircraft measurements. Atmospheric Chemistry & Physics, 16(23).*

**P8L32: the figure does not provide anomaly for the seasonal cycle, so that this statement is poorly supported**

Yes you are right, we removed this sentence from the manuscript.

**P10L18: I find figure S4 more insightful than Figure 6, suggest swapping both**

We moved this figure to the manuscript and added a discussion:

*We can see substantial improvement of the trend estimate when more data are available, with a similar trend accuracy for both approaches. However, in terms of trend precision, the integrated fit produces lower uncertainty.*

**P11L22: which geophysical process could explain the difference vertical correlation depending on the region?**

Thanks for pointing out this issue, we added a discussion on the difference of vertical correlation between Europe and China:

*This vertical correlation range is greater than was found for the IAGOS data above Western Europe. The longer correlation range is the result of the more systematic temporal variations across ozone vertical profiles, as seen in Figure 10(b). While the correlation structure above Europe is heavily affected by the high anomalies from stratospheric influences, if those high anomalies are filtered out (see supplementary Figure S2), the vertical correlation range increases and becomes similar to the IAGOS data above China*

**P12L2: add percentiles in the figure legend**

Figure legend was added.

**P14L5: this sentence seems misplaced**

We revised the sentence as following:

*We provide an example in Supplementary Figure S8 to illustrate the application of this method to the quantification of trends in the lower stratosphere above Hilo, Hawaii.*

**Anonymous Referee #2**
We thank the reviewer for providing valuable comments on our manuscript. The reviewer comments are shown below in bold font, followed by our response in normal font.

**My initial impression and doubts remain. The paper is well written and presented, the discussion accurate although quite technical. Overall scopes are, in my opinion, quite narrow. I still see the manuscript better suited for a journal with a more speciliased focus on the subject of statistical methods, or on measurements.**

**Anyhow, the subject covered here is explicitly mentioned in the ACP subject areas (https://www.atmospheric-chemistry-and-physics.net/about/subject_areas.html) and several references used by the authors confirm that other studies dealing with trend detection and tropospheric measurements have been published in ACP.**

**I therefore advice the editor to accept the manuscript for publication, and suggest the authors to simplify the discussion where possible, to reach out larger audience. Perhaps some technical material can be moved to appendix?**
Thanks for the suggestion, we moved the technical details from Section 2.1 to Appendix A.

**Moreover, what about comparing the result of your methodology with profiles treated with other methods mentioned by the authors in the introduction? this would help determine what is the cost-effectiveness with respect to benefits of your approach compared to simpler ones.**
To our knowledge, previous work on the trend detection of vertical profile data simply averaged the observations into fewer layers (i.e. lower, mid and upper troposphere), or treated each pressure level independently. Therefore this work is the first attempt to obtain a vertically resolved result of trend distribution by a systematic approach (taking vertical correlation into account). The other methods described in the introduction deal with spatial averaging or using principal component analysis, which are very different from trying to take advantage of the correlation between adjacent layers.

A useful comparison between our approach and the standard method of ignoring vertical correlation is demonstrated by the sensitivity analysis of sampling frequency, as we provided in Section 3.2. Nevertheless, we have further demonstrated the benefit of our approach with a new analysis added to Section 3.2, which focuses on sampling strategy:

*In order to investigate the impact of sampling strategy on the integrated and separated fits, we carry out an additional sensitivity analysis in supplementary Table S2 by a comparison of two strategies: (A) a completely random design as illustrated above; (B) a fixed sampling strategy based on the random selection of an initial day followed by additional profiles at fixed intervals of 1 to 10 days. For example, a 5-day sampling frequency is based on the random selection of an initial reference from day 1 to day 5 in the beginning of the record, followed by the random selection of a profile every 5th day until the end of the record. For both strategies, only a single*

*profile is chosen randomly if multiple profiles are available on the same day. Therefore the sampling scheme with a fixed frequency of 1 day represents a random selection of a single profile in each day, instead of using all available data. Also, if the chosen day does not have any profiles, we treat it as missing and do not look for a replacement.*

*The sensitivity analysis demonstrates that sample size remains the dominant influence on precision and accuracy. When the sampling interval is not dense enough (e.g. greater than 10-days), the benefit of a regular frequency scheme is inconsequential. However, when the monthly sample size is greater than 4 profiles (i.e. the sampling frequency is less than once per week), the fixed frequency scheme could achieve a better performance. As a result, an optimal frequency can decrease to 10 profiles (3-day frequency) for an integrated fit and 15 profiles (2-day frequency) for a separated fit.*